# Promoting biomass electrooxidation via modulating proton and oxygen anion deintercalation in hydroxide

Zuyun He[1,4], Jinwoo Hwang [2,4], Zhiheng Gong[1], Mengzhen Zhou[1], Nian Zhang [3], Xiongwu Kang[1], Jeong Woo Han [2✉] & Yan Chen [1✉]

The redox center of transition metal oxides and hydroxides is generally considered to be the metal site. Interestingly, proton and oxygen in the lattice recently are found to be actively involved in the catalytic reactions, and critically determine the reactivity. Herein, taking glycerol electrooxidation reaction as the model reaction, we reveal systematically the impact of proton and oxygen anion (de)intercalation processes on the elementary steps. Combining density functional theory calculations and advanced spectroscopy techniques, we find that doping Co into Ni-hydroxide promotes the deintercalation of proton and oxygen anion from the catalyst surface. The oxygen vacancies formed in NiCo hydroxide during glycerol electrooxidation reaction increase $d$-band filling on Co sites, facilitating the charge transfer from catalyst surface to cleaved molecules during the 2$^{nd}$ C-C bond cleavage. Consequently, NiCo hydroxide exhibits enhanced glycerol electrooxidation activity, with a current density of 100 mA/cm$^2$ at 1.35 V and a formate selectivity of 94.3%.

[1] School of Environment and Energy, State Key Laboratory of Pulp and Paper Engineering, South China University of Technology, 510006 Guangzhou, Guangdong, China. [2] Department of Chemical Engineering, Pohang University Science and Technology, Pohang, Gyeongbuk 37673, Republic of Korea. [3] State Key Laboratory of Functional Materials for Informatics, Shanghai Institute of Microsystem and Information Technology, Chinese Academy of Sciences, 200050 Shanghai, China. [4] These authors contributed equally: Zuyun He, Jinwoo Hwang. ✉email: jwhan@postech.ac.kr; escheny@scut.edu.cn

Electrocatalytic refinery, which converts biomass feedstocks to transportable fuels and value-added chemicals, is regard as a promising substitute for traditional fossil fuel-based industrial refineries because of its sustainable and carbon-neutral nature[1]. In addition, the electrocatalytic refinery is decentralized and independent of large-scale equipment, making it suitable for on-demand production[2,3]. Biomass electrooxidation reaction (BEOR) occurred on the anode is one critical reaction in electrocatalytic refinery. Despite many pioneering works on developing highly active and stable electrocatalyst for BEOR[4–8], the correlation between material properties and reaction kinetics is still ambiguous, largely due to the complex reaction pathway, involving reaction steps such as dehydrogenation, adsorption/ desorption of reaction intermediates, and C–C bonds cleavage.

One representative reaction for BEOR is the glycerol oxidation reaction (GOR). As a byproduct from biodiesel production, glycerol is listed as one of the top ten biomass-derived platform molecules for value-added chemicals production by the U.S. Department of Energy[9,10]. Various products were reported in electrooxidation of glycerol, such as glyceraldehyde[11], dihydroxyacetone[12–15], glyceric acid[16–19], glycolic acid[20], and formic acid[4,21–28]. As a consequence of such diversity in products and complexity in reaction pathway, controlling reaction pathway during GOR remains challenging, leading to a poor selectivity toward the desired products. It is essential to identify the key material characteristics which determine the elementary steps of GOR on the catalyst surface.

The redox center of transition metal oxides and hydroxides is generally considered to be the metal site[29]. Tuning properties related to metal sites such as the electro-positivity or oxidation state was widely used for improving the electrocatalytic activity[30]. Interestingly, proton (de)intercalation process recently was reported to be of great significance to the electrochemical catalytic reaction[31–36]. For instance, Zhu et al.[31] tuned the proton diffusion in $PrBa_{0.5}Sr_{0.5}Co_{1.5}Fe_{0.5}O_{5+\delta}$ double perovskite oxide by crystal orientation, leading to improved proton-coupled electron transfer process and enhanced oxygen evolution reaction (OER) activity. Tse et al.[36] regulated proton transfer kinetics of Cu-based catalyst by lipid modification to effectively promote oxygen reduction reaction. In addition to proton, the (de)intercalation of oxygen anion from lattice was also reported to critically influence the activity of electrocatalyst[37–46]. For example, Grimaud et al.[43] and Huang et al.[45] observed that lattice oxygen in perovskite oxide actively participates in the surface OER. Chen et al.[44] tuned the oxygen activity, which was represented by the metal–oxygen bond strength and oxygen vacancy formation energy, in perovskite ferrite by Co doping, and observed strongly facilitated hydrogen oxidation reaction at elevated temperature. While all these works suggest that regulating the proton and oxygen anion (de)intercalation in metal oxide/hydroxide is a potential way to improve the electroactivity, very limited work was carried out for BEOR, and the detailed mechanism is still not fully understood.

In this work, we reveal systematically how proton and oxygen anion (de)intercalation processes in hydroxide catalysts determine the elementary reaction steps in GOR by using the combination of advanced spectroscopy techniques and density functional theory (DFT) calculations. We find that the glycerol dehydrogenation reaction step is spontaneous on Ni, NiCo and Co hydroxide, and strongly correlates to the electrochemical driven deintercalation of proton from hydroxide lattice. In addition, the desorption of reaction products is accompanied by the oxygen anion deintercalation, leading to oxygen vacancy formation on the electrocatalyst surface. The oxygen vacancies formed on the NiCo hydroxide surface during GOR increase the $d$-band filling on Co site, which facilitates the charge transfer from the catalyst surface to the cleaved molecules and promotes

the $2^{nd}$ C–C bond cleavage step. As a result, NiCo hydroxide exhibits enhanced glycerol electrooxidation activity, requiring only 1.35 V to reach 100 mA/cm$^2$ and with formic acid selectivity of 94.3%. Our work clarifies the critical role of proton and oxygen deintercalation processes on the elementary steps in biomass electrooxidation, proving guideline for further design of high-performance catalysts.

## Results

**Electrocatalytic performance towards glycerol oxidation.** Ni hydroxide was synthesized on carbon cloths by electrodeposition, and Co doping was introduced by cation exchange reaction (Supplementary Fig. 1). The detailed structure and chemical composition of Ni hydroxide, NiCo hydroxide, and Co hydroxide reference samples were shown in Supplementary Note 1 and Supplementary Figs. 2–11. The atom ratio of Ni to Co in NiCo hydroxide was identified to be 1.26.

The GOR performances of Ni, NiCo and Co hydroxide were evaluated in a double-compartment H-cell with typical three-electrode configuration. As shown in the cyclic voltammetry (CV) curves (Fig. 1a, Supplementary Fig. 12), all the samples exhibited much higher selectivity towards the glycerol oxidation (solid line) than water oxidation (dash line). The NiCo hydroxide delivered enhanced GOR activity, requiring only 1.35 V (vs. RHE) to reach a current density of 100 mA/cm$^2$. The highest GOR performances of NiCo hydroxide were further confirmed by the Tafel curves and electrochemical impedance spectroscopies (EIS) (Fig. 1b, c). The EIS spectra of all samples exhibited a characteristic semicircle shape, which were fitted by an Ohmic resistance ($R_s$), a constant phase element (CPE), and a charge-transfer resistance ($R_{ct}$) (Supplementary Fig. 13). NiCo hydroxide exhibited the smallest charge-transfer resistance value among all samples, indicating the fastest GOR kinetics. To rule out the impacts of microstructure, the current density was also normalized by electrochemical surface area (ECSA), which was estimated by double-layer capacitance ($C_{dl}$) (Supplementary Figs. 14 and 15). After ECSA normalization, NiCo hydroxide still exhibited the optimal GOR performance among all samples. Turnover frequency results provided the same conclusion (Supplementary Note 2, Supplementary Fig. 16). In addition, NiCo hydroxide exhibited a higher mass activity than Ni hydroxide (Supplementary Fig. 17), suggesting that Co doping can effectively enhance the intrinsic activity of Ni hydroxide for GOR. These results suggested that NiCo hydroxide was indeed intrinsically more active toward GOR than Ni and Co hydroxide.

The higher GOR activity of NiCo hydroxide than Co and Ni hydroxide was also supported by the quantification of the reaction products of glycerol oxidation. As shown in the high-performance liquid chromatography (HPLC) results, the reaction products mainly contained glycerate, glycolate, and formate (Supplementary Figs. 19 and 20). For all samples, formate was detected as the main product, while glycerate and glycolate were detected as the minor products (Supplementary Fig. 21).

We also collected the reaction products online during the LSV measurement and identified its composition by ex-situ HPLC measurement (Supplementary Fig. 22). As the applied potential increased, the concentration of formate showed a significant increase, while the concentration of glycerate and glycolate remained unchanged. This result suggested that formate is the main product, which was consistent with the HPLC results after long-time electrolysis.

The formate production rate exhibited a trend of NiCo > Ni > Co hydroxide (Supplementary Fig. 23), indicating the fastest GOR kinetics of NiCo hydroxide. This result is consistent with the one we observed in electrochemical measurement (Fig. 1a–c).

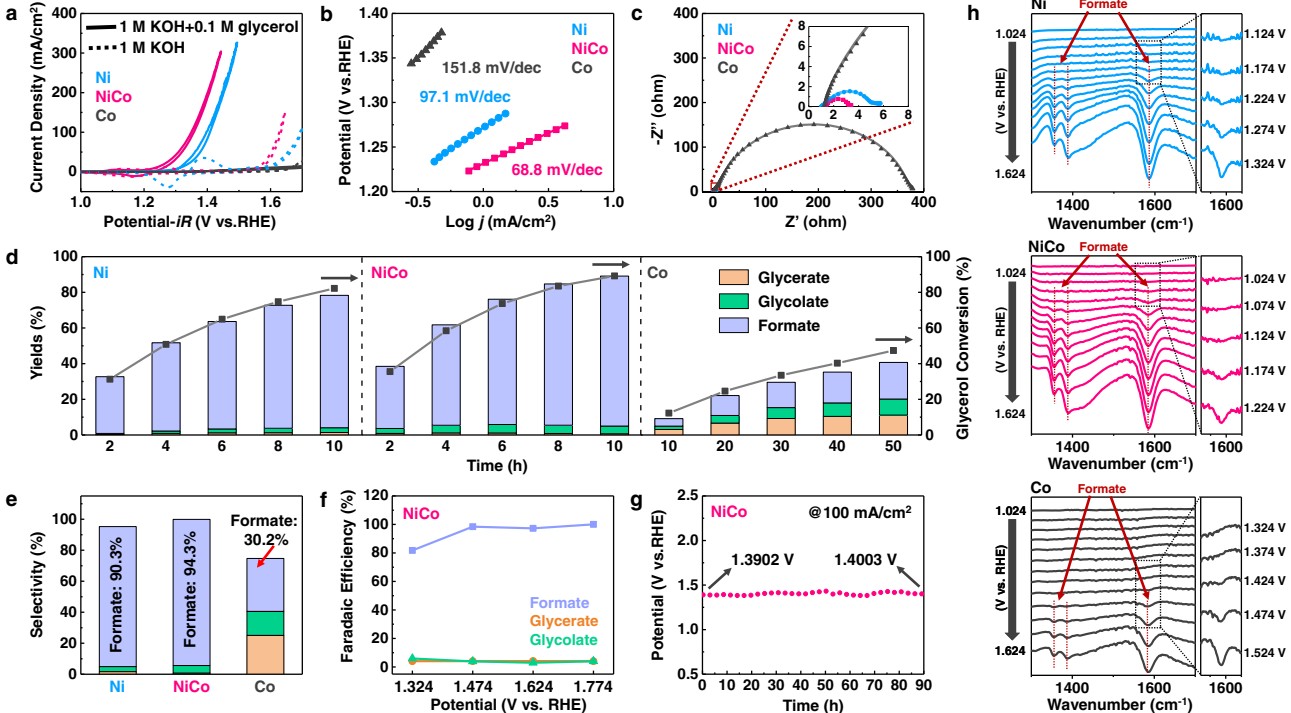

**Fig. 1 Evaluation of the electrocatalytic performance towards glycerol oxidation reaction (GOR). a** Cyclic voltammetry curves of Ni, NiCo, and Co hydroxide in 1 M KOH solution with (solid line) or without (dash line) 0.1 M glycerol. **b** Tafel curves and **c**, electrochemical impedance spectroscopies of Ni, NiCo, and Co hydroxide for GOR. The EIS measurements were performed at the potential of 1.324 V (vs. RHE). **d** The evolution of glycerol conversion and product yields as a function of electrolysis time at the potential of 1.624 V (vs. RHE) in 50 mL electrolyte of 1 M KOH with 0.1 M glycerol. The corresponding i–t curves were shown in Supplementary Fig. 18. **e** The selectivity of different products after electrolyzing at 1.624 V (vs. RHE) for 10 h in 50 mL electrolyte of 1 M KOH with 0.1 M glycerol. **f** The evolution of Faradaic efficiency as a function of applied potential on NiCo hydroxide with the same amount of total charge passed (1000 C). **g** Chronopotentiometry curve for NiCo hydroxide acquired at the current density of 100 mA/cm². **h** In-situ infrared spectra of Ni, NiCo, and Co hydroxide obtained during the linear sweep voltammetry (LSV) measurement. The right panels are the enlarged figures of the dash line areas.

The glycerol conversion and product yields were calculated based on carbon balance (Supplementary Note 3). As shown in Fig. 1d, NiCo hydroxide exhibited the highest glycerol conversion rate and product yields rate. NiCo hydroxide showed a formate selectivity of 94.3%, which was better than 90.3% for Ni hydroxide (Fig. 1e). Co hydroxide showed the most diverse products with a large amount of glycerate and glycolate. The Faradaic efficiency of NiCo hydroxide for formate production was close to 100% when the applied potentials were higher than 1.474 V (vs. RHE) (Fig. 1f, Supplementary Fig. 24). NiCo hydroxide also exhibited good stability for GOR, with negligible changes in the potential during the chronopotentiometry measurement for 90 h at a constant current density of 100 mA/cm² (Fig. 1g).

We further compared the kinetics of GOR on different samples by monitoring the formation of formate on catalyst surface during the linear sweep voltammetry (LSV) measurement using in-situ infrared spectroscopy technique (Fig. 1h). As the applied potential increased, the absorption bands located at 1356, 1388, 1585 cm⁻¹ appeared on all three samples, which were corresponding to the C–O symmetric stretching, COO rocking, and C–O asymmetric stretching of formate, respectively[47,48]. NiCo hydroxide sample required only 1.174 V (vs. RHE) for formate formation, which was much lower than the 1.274 V (vs. RHE) for Ni hydroxide and 1.474 V (vs. RHE) for Co hydroxide, respectively. This result was consistent with the electrochemical test and the products analysis, suggesting that NiCo hydroxide showed the highest activity towards the conversion of glycerol to formate. However, the corresponding absorption bands of

glycerate (1580 and 1419 cm⁻¹) and glycolate (1580, 1410, 1326, and 1075 cm⁻¹) cannot be observed due to their low concentration. In addition, the absence of the characteristic broad band of carbonate at around 1400 cm⁻¹ suggested that formate could remain stable without further oxidation to form carbonate, which was consistent with the almost 100% of Faradic efficiency for formate production.

**Revealing glycerol oxidation mechanism.** Having confirmed that glycerate, glycolate, and formate are the main reaction products of GOR on Ni, NiCo, and Co hydroxide, we then carried out DFT calculations to determine detailed reaction pathways (Supplementary Figs. 25–28, Supplementary Note 4 and 5). As shown in Fig. 2a, the proton and oxygen anion (de)intercalation in hydroxide were found to be actively involved in the elementary reaction steps, including the initial electrochemical driven deintercalation of proton from electrocatalyst lattice (IS → 0) and the oxygen anion deintercalation during the desorption of the reaction products (3 → A1 → A2, 7 A → 7B, B1 → B2 and 9B → FS).

The Gibbs free energy diagrams of GOR on Ni, NiCo, and Co hydroxides are shown in Fig. 2b, Supplementary Table 2, and Supplementary Note 6. The dehydrogenation process of the adsorbed reaction intermediates (1 → 2, 2 → 3, 4 → 5) were found to be spontaneous with negative changes in Gibbs free energies. Particularly, the intermediates on Co hydroxide showed the most negative dehydrogenation energy in comparison to that on NiCo or Ni hydroxide. The protons detached from the reaction intermediate during the dehydrogenation reaction went

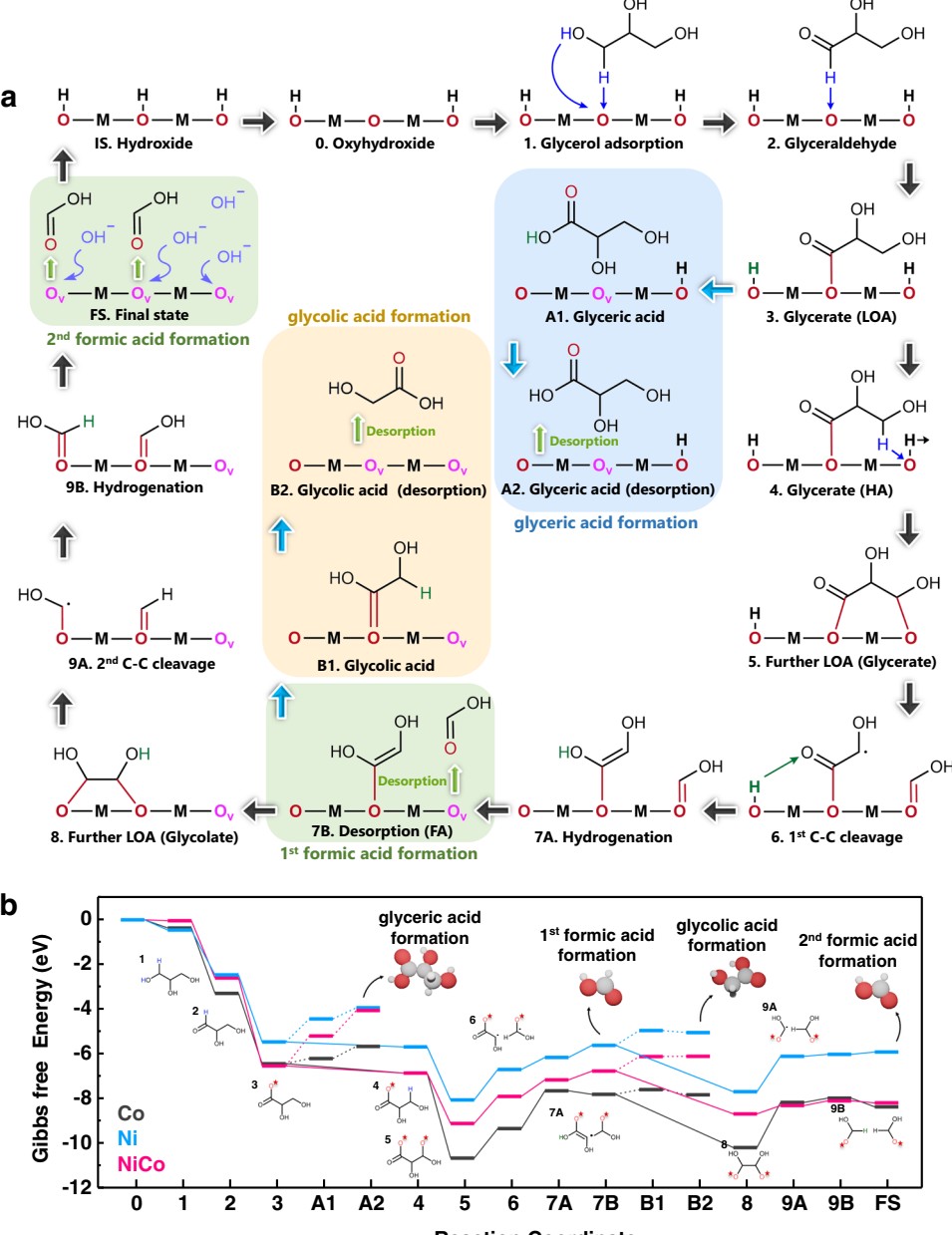

**Fig. 2 Theoretical determination of the glycerol oxidation reaction pathway. a** Schematic illustration of the glycerol oxidation reaction pathway. The corresponding configurations are shown in Supplementary Figs. 26–28. **b** The calculated Gibbs free energy profiles of glycerol oxidation reaction on Ni, NiCo, and Co hydroxide. The numerical data are shown in Supplementary Table 2.

directly to the catalyst surface to intercalate into the –OOH site and recover the –OH surface (Supplementary Figs. 26–28).

All the desorption of the reaction products ($3 \rightarrow$ A1 $\rightarrow$ A2, 7 A $\rightarrow$ 7B, B1 $\rightarrow$ B2 and 9B $\rightarrow$ FS) were found to involve the oxygen anion deintercalation process, leading to oxygen vacancies formation on the surface. Co hydroxide exhibited the smallest barrier for all these desorption reaction steps. As will be shown in the later section, such low desorption barrier was related to the low oxygen vacancy formation energy of Co hydroxides. Because the reaction products were very easy to desorb from surface, Co hydroxide showed very diverse products, with a large amount of glycerate and glycolate in addition to formate (Fig. 1e).

The C–C bond cleavage steps were found to be with high-energy barrier for all three samples. While all samples showed similar barrier for the 1st C–C bond cleavage ($5 \rightarrow 6$), NiCo hydroxide showed the lowest barrier in the 2nd C–C bond

cleavage step ($8 \rightarrow 9$ A). Such low barrier for the 2nd C–C bond cleavage in NiCo hydroxide facilitateed the formation of formic acid as the final reaction product. This theoretical result was consistent with the extra high efficiency and selectivity for formate production that we observed experimentally on NiCo hydroxide.

In addition, the reaction energy calculations were also performed (Supplementary Fig. 30, and Supplementary Table 3), which showed consistent results with the calculated Gibbs free energy in terms of catalytic activity trend.

**Impact of (de)intercalation of proton in hydroxides.** As predicted by the DFT calculation, the proton deintercalation from hydroxide lattice occurred before the GOR (IS $\rightarrow$ 0), and the dehydrogenation of the adsorbed intermediate on the oxyhydroxide surface was spontaneous. To confirm such theoretical

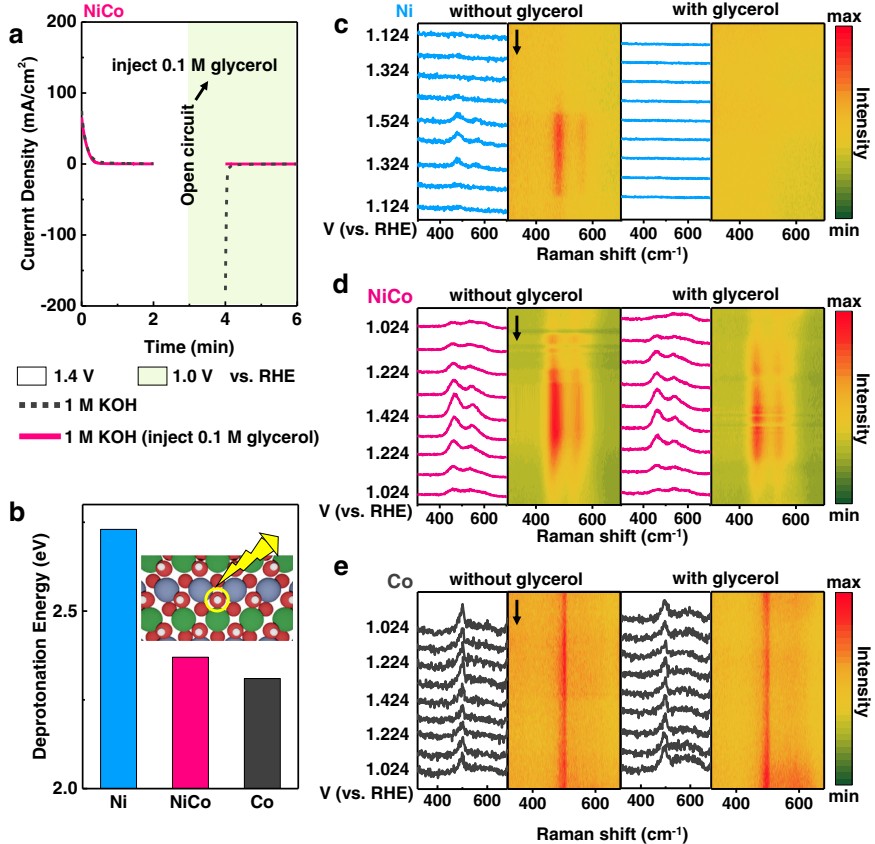

**Fig. 3 The impact of proton deintercalation process on the dehydrogenation step during GOR. a** Muti-potential step curves of NiCo hydroxide.
**b** Deprotonation energy, i.e, energy required for the deintercalation of one proton from the lattice, of Ni, NiCo, and Co hydroxide. The green, blue, red, and white balls represent the Ni, Co, O, and H atoms, respectively. In-situ Raman spectroscopies for **c** Ni hydroxide, **d** NiCo hydroxide, and **e** Co hydroxide in 1 M KOH without (left) and with (right) 0.1 M glycerol.

results, we carried out an intermittent GOR measurement by applying different potentials on the electrocatalyst and monitoring the changes in the electrochemical current (Fig. 3a and Supplementary Fig. 31). Such method was reported to be capable of separating the process of proton deintercalation from the catalyst lattice and the dehydrogenation reaction of the biomass during the electrooxidation reaction[32,33].

An oxidation current was observed when an anodic potential of 1.4 V (vs. RHE) was first applied on NiCo hydroxide with 1.0 M KOH as the electrolyte. This oxidation current was related to the electrochemical driven deintercalation of proton from hydroxide lattice to form oxyhydroxide (−OH → −OOH)[32,33]. The applied anodic potential was then withdrawn and the glycerol was injected during the open circuit state. Subsequently, a cathodic potential of 1.0 V (vs. RHE) was applied on the catalyst and we observed no reduction current (Fig. 3a, solid line). By contrast, a large reduction current was observed when no glycerol was added during the open circuit state (Fig. 3a, dash line). This result indicated that the electrogenerated NiCo oxyhydroxide can accept protons from glycerol at open circuit state during glycerol dehydrogenation step and then transform to hydroxide spontaneously, thus reduction current cannot be detected. Similar phenomena were observed for Ni and Co hydroxide (Supplementary Fig. 31). Our experimental results implied that the glycerol dehydrogenation reaction on Ni, NiCo, and Co oxyhydroxide is spontaneous, which was consistent with the DFT calculation results shown in the previous section.

Because of the spontaneous nature of the glycerol dehydrogenation reaction on the −OOH surface (Figs. 2b and 3a), the

glycerol dehydrogenation reaction rate could be strongly impacted by the electrochemical driven deintercalation of proton from hydroxide lattice (oxyhydroxide formation). The deprotonation energy (energy required for the deintercalation of one proton from the lattice) of Ni, NiCo, and Co hydroxide were determined by DFT calculation (Fig. 3b), with a trend of Ni (2.73 eV) > NiCo (2.37 eV) > Co (2.31 eV). The lower deprotonation energy of NiCo hydroxide than Ni hydroxide suggested that Co doping in Ni(OH)$_2$ could effectively promote the deintercalation of proton from hydroxide to form oxyhydroxide. Such effects of Co doping could also be confirmed by the negative shift of the redox peak after Co doping as we observed in the CV measurement in KOH (Supplementary Fig. 32).

As shown in the previous DFT calculation results, protons generated from the glycerol dehydrogenation reaction would go to the surface of hydroxides and intercalate into the oxyhydroxide (−OOH) site. As a consequence, the competing effect of the proton deintercalation from hydroxide lattice and the glycerol dehydrogenation process determined the amount of oxyhydroxide (−OOH) on the surface, which could be revealed by the in-situ Raman measurements (Fig. 3c–e). For Ni hydroxide, two bands at 481 and 567 cm$^{-1}$ were observed in 1 M KOH when an anodic potential was applied, which were attributed to $e_g$ bending and $A_{1g}$ stretching vibration mode of Ni$^{3+}$−O in NiOOH (Fig. 3c left)[49–51]. Such Raman bands for NiOOH did not appear throughout the electrochemical potential window when measuring in 1 M KOH with 0.1 M glycerol (Fig. 3c right). This result indicated that NiOOH could not accumulate on the surface during GOR, and the GOR on Ni hydroxide was limited by the

oxyhydroxide formation (Supplementary Fig. 33a). By contrast, for NiCo and Co hydroxide (Fig. 3d, e), Raman bands assigned to $Ni^{3+}$–O or $Co^{3+}$–O were observed when testing both in KOH with/without glycerol. These results suggested that the proton deintercalation from the lattice of NiCo and Co hydroxide occured more rapidly than the glycerol dehydrogenation reaction during GOR (Supplementary Fig. 33b).

The intermittent GOR test and in-situ Raman measurement demonstrated that the GOR on Ni hydroxide was strongly limited by the proton deintercalation from catalyst surface, which was likely due to its high deprotonation energy. Doping Co into Ni hydroxide could effectively promote the proton deintercalation, facilitating the dehydrogenation step in GOR. In addition, such promoted proton deintercalation could expose more lattice oxygen sites for the further lattice oxygen attack step ($2 \rightarrow 3$, $4 \rightarrow 5$, $7B \rightarrow 8$, Supplementary Note 4).

It is noted that Co hydroxide exhibited the lowest deprotonation energy. Consistently, Co hydroxide showed Co–OOH Raman peak throughout whole electrochemical window (1.024–1.424 V vs. RHE) (Fig. 3e). Nevertheless, Co hydroxide showed the worst GOR activity, implying that the deintercalation of protons from hydroxide was not the only factor for determining the GOR activity.

**Impact of (de)intercalation of oxygen anion in hydroxides.** As revealed in the reaction pathway (Fig. 2a), the desorption of products was accompanied by oxygen anion deintercalation from hydroxide lattice (i.e. oxygen vacancy formation). Therefore, it was expected that the ease of oxygen anion deintercalation from the lattice was critical to the desorption process. Using DFT calculation, the energy required for the deintercalation of one oxygen from the lattice (formation energy of a single oxygen vacancy, $E_{f\_vac}$) in Ni, NiCo, and Co hydroxide was calculated to be 1.28, 1.05, and −0.59 eV, respectively. Taking the desorption of formic acid after first C–C cleavage ($7A \rightarrow 7B$) as the example, we found that the energy barrier of the desorption step was strongly correlated with the $E_{f\_vac}$ value (Fig. 4a). Similar correlation was also found between the reaction barrier and the energy required for the deintercalation of two or three oxygen anions from the lattice (formation energy of double or triple oxygen vacancies) (Supplementary Fig. 34). Co hydroxide with the lowest energy for oxygen anion deintercalation exhibited the lowest barrier for the products to desorb from the catalyst surface. As a consequence, the reaction products tended to leave the surface after formation instead of proceeding further oxidation, which resulted in the poor selectivity of Co hydroxide for formate production (Fig. 1e). Ni and NiCo hydroxides had relatively high energy for oxygen anion deintercalation, which facilitated the further oxidation of glycerol for the final production of formate.

Oxygen vacancies formed during GOR was further found to critically impact the local electronic structure and determine the subsequent oxidation process. As predicted by DFT calculation, after oxygen anion deintercalation from the lattice (oxygen vacancy formation), electrons transferred away from the Ni sites on Ni hydroxide and NiCo hydroxide, leading to a decreased $d$-band filling of Ni sites (Fig. 4b). By contrast, oxygen anion deintercalation caused no changes in the $d$-band filling of Co sites on Co hydroxide, but led to an increased $d$-band filling of Co sites on NiCo hydroxide (Fig. 4b). The same conclusion about the electron redistribution induced by oxygen anion deintercalation could also be drawn from density of states (DOS) and crystal orbital Hamilton population (COHP) calculations (Supplementary Figs. 35–38, Supplementary Note 7).

To confirm these changes in electron filling predicted by DFT, we carried out soft X-ray absorption spectroscopy (sXAS)

measurements on the sample after operating at 1.624 V (vs. RHE) in KOH with/without glycerol, which represented the sample states with/without oxygen vacancy. The unoccupied states in $d$-orbitals of transition metals were probed by metal L-edge sXAS spectra, which derived from dipole-allowed $p$ to $d$ electron transition[30]. For the Ni and NiCo hydroxides reacting with glycerol, the Ni L-edge located at higher photo energy than the one reacting with only KOH (Fig. 4c). Such difference suggested that the Ni in Ni and NiCo hydroxide present higher oxidation state[52], i.e. lower $d$-band electron filling, after reacting with glycerol and forming oxygen vacancy. This result was consistent with the lower $d$-band filling on Ni site of Ni hydroxide predicted by DFT calculation (Fig. 4b). In the Co L-edge spectra, an extra peak of $Co^{2+}$ appeared in the pre-edge region for the NiCo hydroxide after reacting with glycerol, suggesting a lower oxidation state of cobalt. By contrast, the Co hydroxide after reacting with KOH with/without glycerol showed the same Co L-edge spectra. These changes in Co L-edge spectra were consistent with changes in $d$-band filling of Co sites on NiCo hydroxide and Co hydroxide after oxygen anion deintercalation (Fig. 4b).

Having confirmed by DFT calculation and sXAS spectra that oxygen anion deintercalation during GOR lead to noticeable changes in the electron filling on the metal sites, we will further show the impact of such changes in electron filling on the subsequent $2^{nd}$ C–C bond cleavage, which is the rate-limiting step for formic acid production.

The reaction barrier for the $2^{nd}$ C–C bond cleavage ($8 \rightarrow 9A$) was plotted together with the $d$-band filling of metal sites after oxygen anion deintercalation (Fig. 4e). It was found that, as more electrons were accumulated on the metal sites (higher $d$-band filling), the $2^{nd}$ C–C bond cleavage was strongly facilitated with lower reaction barrier. The NiCo hydroxide was found to have highest $d$-band filling at both Ni and Co sites, corresponding to its lowest $2^{nd}$ C–C bond cleavage barrier. Similar correlation also found between activation energy and $d$-band filling (Supplementary Figs. 39, 40, Supplementary Note 8). In addition, the bader charge analysis was performed on the cleaved molecule on the sample surface after the $2^{nd}$ C–C bond cleavage (Fig. 4f). The charge on cleaved molecules showed a trend of NiCo > Ni > Co hydroxide, which was strongly correlated to the $d$-band filling on metal sites. The NiCo hydroxide exhibited the most charge transfer to the cleaved molecules, which led to a noticeable upshift of the band center of Co $3d$ in NiCo hydroxide after $2^{nd}$ C–C cleavage as revealed by DOS calculation (Fig. 4g). According to $d$-band theory, metal sites with higher $d$-band center possess a higher bonding strength for the cleaved molecules (oxygenated intermediates) owing to the reduced electron filling of the anti-bonding states[53–57]. This result was consistent with the lowest adsorption energy and highest stability of cleaved molecules on NiCo hydroxide with the highest $d$-band filling (Supplementary Figs. 41 and 42, Supplementary Note 9). Corresponding to the $d$-band upshift during the C–C bond cleavage process, Co sites on NiCo hydroxide devoted electrons, resulting in the decrease of $d$-band filling from 0.90 to 0.57 (Supplementary Fig. 43). In contrast to Co sites, the $d$-band filling did not change much on the Ni sites. These results suggested that the Co sites in NiCo hydroxide served as the active sites for the $2^{nd}$ C–C bond cleavage (Supplementary Note 10).

On the basis of all the results above, we believe that, during oxygen anion deintercalation from the lattice, electrons accumulating on Co sites of NiCo hydroxide, hence more charges should be transferred to the cleaved molecule and stabilize the molecules on the surface during the $2^{nd}$ C–C bond cleavage. As a consequence, NiCo hydroxides exhibited the lowest energy barrier on the $2^{nd}$ C–C bond cleavage, which is the rate limiting step for formic acid production.

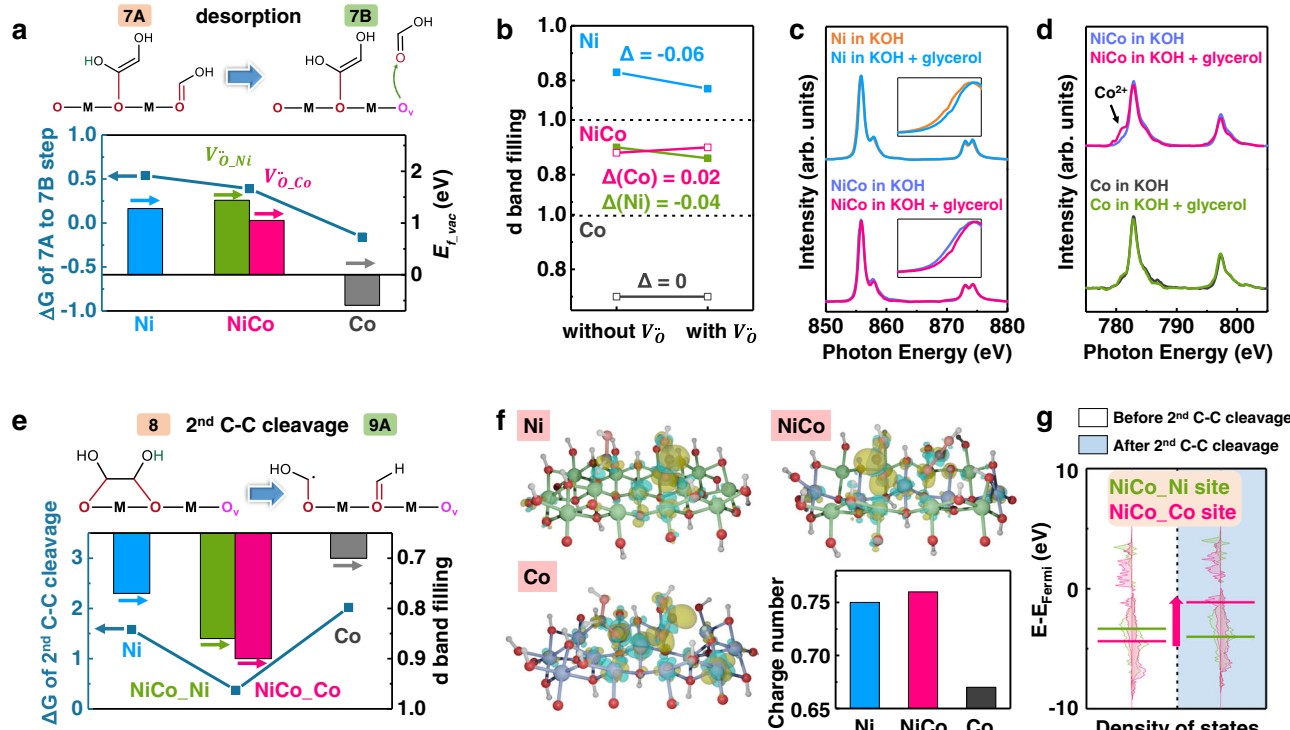

**Fig. 4 The impact of oxygen anion deintercalation process on the desorption step and 2nd C–C bond cleavage step. a** The free energy barrier of the desorption step (7A → 7B) and the energy for the deintercalation of one oxygen anion from the hydroxide lattice ($E_{f\_vac}$) for Ni, NiCo, and Co hydroxide. **b** The change in $d$-band filling of the metal sites before and after oxygen anion deintercalation from the lattice. $V_{\ddot{O}}$ is referred to oxygen vacancy. **c** The Ni L-edge and **d** Co L-edge soft X-ray absorption spectra (sXAS) of Ni, NiCo, and Co hydroxide after operating at 1.624 V (vs. RHE) in 1 M KOH with/without 0.1 M glycerol. The insets in **c** are the corresponding magnified images of the pre-edge. The area of sXAS spectra were normalized to be 1. **e** The free energy barrier of the 2nd C–C bond cleavage step (8 → 9A) and the $d$-band filling before 2nd C–C bond cleavage of Ni, NiCo, and Co hydroxide. NiCo_Ni and NiCo_Co represent the Ni sites and Co sites on NiCo hydroxide, respectively. **f** The charge density difference plots and the bader charge analysis of the cleaved molecule on Ni, NiCo, and Co hydroxide after the 2nd C–C bond cleavage. The yellow area indicates the charge accumulation, while cyan area indicates the charge depletion. **g** The DOS of NiCo hydroxide before (white area) and after (green area) the 2nd C–C bond cleavage. The lines indicate the corresponding band center.

**Overall electrolysis coupling with hydrogen evolution reaction.** To further demonstrate the application of hydroxide as highly active catalysts for electrocatalytic refinery, we constructed an asymmetric electrolytic cell coupling GOR with hydrogen evolution reaction (HER). Similar to GOR activity, the NiCo hydroxide also exhibited the optimal HER activity among all samples (Supplementary Fig. 45). The cell with NiCo hydroxide as electrodes for both GOR and HER exhibited superb overall electrolysis performance, requiring only 1.33 and 1.58 V to reach a current density of 10 and 100 mA/cm², respectively (Fig. 5a). After normalizing the current density by the double-layer capacitance ($C_{dl}$) (Supplementary Fig. 46), NiCo hydroxide achieved a current density of 11.96 A/F at the voltage of 1.5 V, which was much higher than that of Ni hydroxide (4.32 A/F) and Co hydroxide (1.20 A/F), suggesting that NiCo hydroxide indeed exhibited a much higher activity than the Ni hydroxide and Co hydroxide for overall electrolysis coupling GOR with HER. Such performance was competitive among the noble-metal-free electrocatalysts reported in the literature for overall electrolysis by coupling organic oxidation reactions with hydrogen production (Fig. 5b, Supplementary Table 4). In addition, the electrolytic cell with NiCo hydroxide electrodes also showed good stability, with only 27.5 mV increasement in voltage after 110 h electrolyzing at the current density of 100 mA/cm² (Fig. 5c).

## Discussion

In summary, we employed both computational and experimental approaches to reveal how proton and oxygen anion (de)

intercalation in hydroxide impact the elementary reaction step of the glycerol oxidation reaction. We found that the ease of proton deintercalation from hydroxide lattice critically impacted the dehydrogenation step in GOR, while the oxygen anion deintercalation process determined the product desorption step and final reaction selectivity. Oxygen vacancies formed in NiCo hydroxide during GOR increased $d$-band filling of Co sites, promoting the subsequent 2nd C–C bond cleavage, which was the rate-limiting step for formic acid production. Owing to the facilitated proton and oxygen (de)intercalation process, NiCo hydroxide exhibited enhanced GOR activity, delivering a current density of 100 mA/cm² at 1.35 V with a formate selectivity of 94.3%. Coupling GOR with hydrogen production, NiCo hydroxide achieved a current density of 10 mA/cm² at 1.33 V. Such understanding about the critical impact of proton and oxygen anion (de)intercalation processes on biomass electrooxidation activity can guide the design of high-performance electrocatalysts for electrocatalytic refinery.

## Methods

**Synthesis of catalysts**. Ni hydroxide was prepared on carbon cloths by electrodeposition. Prior to the electrodeposition, the carbon cloths were treated by 37% hydrochloric acid at 100 °C for 2 h and dried at 60 °C for 5 h in vacuum. The electrodeposition was performed in a three-electrode system using a CHI-660E electrochemical station. 0.1 M nickel acetate solution was used as electrolyte and the treated carbon cloths acted as the working electrode. A Ag/AgCl electrode prefilled with saturated KCl aqueous solution and a Pt mesh were used as reference electrode and counter electrode, respectively. A constant cathodic current density of 10 mA/cm² was exerted on working electrode by chronopotentiometry for 5 min to obtain Ni hydroxide nanosheets coating on carbon cloths. Then the resulting

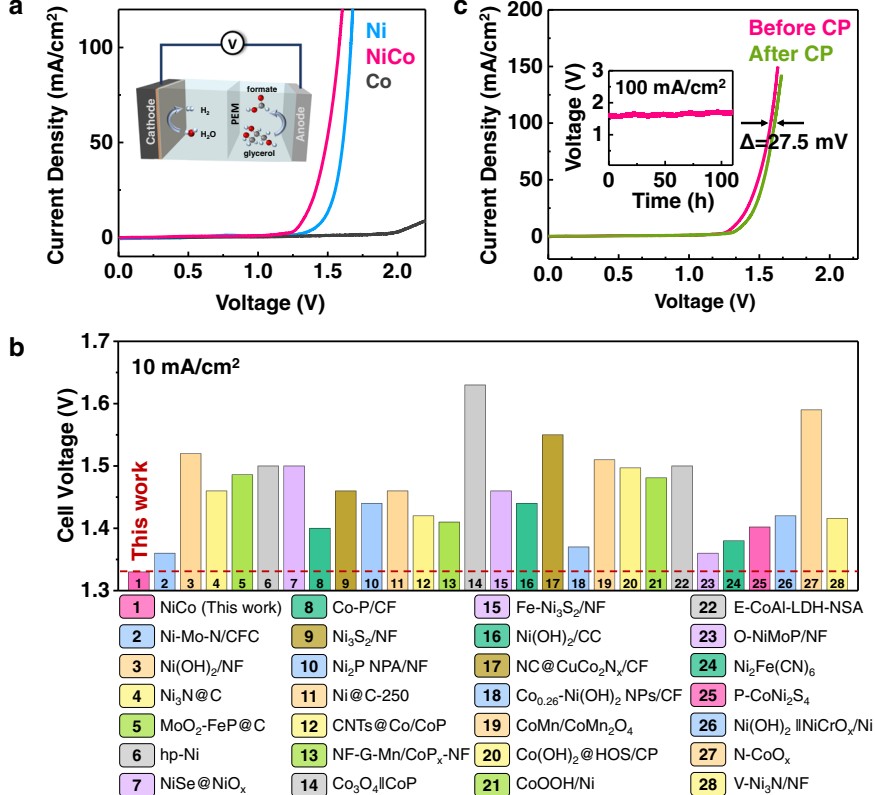

**Fig. 5 Overall electrolysis performance of coupling GOR with hydrogen evolution reaction (HER). a** LSV polarization curves for overall electrolysis coupling HER and GOR. The inset is the schematic illustration of the overall electrolysis configuration. **b** The comparison of overall electrolysis performance coupling organic oxidation reactions with HER in aqueous media at the current density of 10 mA/cm² for NiCo hydroxide and other noble-metal-free electrocatalysts in recently reported literature. **c** LSV polarization curves of NiCo hydroxide before and after chronopotentiometry measurement performed at 100 mA/cm² for 110 h. The inset is the chronopotentiometry curve of NiCo hydroxide performed at 100 mA/cm² for 110 h.

samples were further immersed in 0.1 M cobaltous acetate solution and heated to 80 °C for 2 h to obtain NiCo hydroxide. Then the samples were washed by deionized water thoroughly to remove excess adsorbate and dried at ambient condition for the following characterizations and electrochemical measurements. The loading of metal in NiCo hydroxide was determined to be 0.84 mg/cm² by ICP measurement. Bare-treated carbon cloths went through the same processes except for electrodeposition to obtain Co hydroxide for comparison. More detailed information about chemicals and materials can be found in Supplementary Note 11.

**Characterization**. The morphologies of samples were characterized by high-resolution field emission scanning electron microscopy (SEM) (SU8010, Hitachi, Japan). The structure of catalysts was analyzed by X-ray diffraction (XRD) (D8 Advance, Bruker, Germany) and high-resolution transmission electron microscopy (HRTEM) (JEM-3200FS, JEOL). The chemical composition was detected by X-ray photoelectron spectroscopy (XPS) (Escalab250Xi, Thermo Scientific) and inductively coupled plasma-optical emission spectrometry (ICP-OES) (Agilent 730 series). The in-situ Raman measurements were performed on a confocal microscopic system (LabRAM HR Evolution, Horiba, France) equipped with a semiconductor laser ($\lambda = 532$ nm, Laser Quantum Ltd.). The in-situ infrared spectroscopy measurements were carried out on a Nicolet 6700 FTIR spectrometer (Thermo Scientific, USA) equipped with a liquid-nitrogen-cooled MCT-A detector. sXAS was carried out at the BL02B02 station in Shanghai Synchrotron Radiation Facility[58].

**Electrocatalytic measurement**. The electrochemical measurements were performed in a double-compartment H-cell with typical three-electrode configuration at room temperature. A piece of proton exchange membrane (Nafion 117) was used as a separator. The catalyst-loaded carbon cloth was used as the working electrode. For GOR and OER, a Ag/AgCl electrode prefilled with saturated KCl aqueous solution and a Pt mesh were used as reference electrode and counter electrode, respectively. For HER, a graphite rod was used as reference electrode. 1 M KOH solution with 0.1 M glycerol was used as electrolyte for GOR, while 1 M KOH solution was used as electrolyte for OER and HER. Cyclic voltammetry (CV) and Linear sweep voltammetry (LSV) curves were performed with a scan rate of 5 mV/s. Electrochemical impedance spectra (EIS) were measured at the potential of 1.324 V ($-0.15$ V, vs. RHE) with an amplitude of 5 mV for GOR (HER). The Tafel curves were obtained with a scan rate of 0.1 mV/s. The electrochemical surface area

(ECSA) was evaluated based on the double-layer capacitor, which was obtained from the CV curves in the potential range of 1.024–1.124 V (vs. RHE) with different rates from 100 to 500 mV/s. The overall electrolysis was performed in a two-electrode configuration and the catalyst-loaded carbon cloths were used as both cathode and anode. Chronopotentiometry (CP) tests were carried out at a constant current density of 100 mA/cm² to evaluate the stability of electrolytic cell during long-term operation for glycerol oxidation and overall electrolysis coupling with HER. All given potentials in the manuscript have been converted to be the ones versus reversible hydrogen electrode (RHE). The potentials in CV, LSV, and CP curves in this study have subjected to iR compensation.

**Product analysis**. The glycerol oxidation products were determined by high-performance liquid chromatography (HPLC). Long-term electrolysis reactions of glycerol oxidation were carried out at a constant potential of 1.624 V (vs. RHE) by chronoamperometry in 50 mL electrolyte of 1 M KOH with 0.1 M glycerol. 1 mL electrolyte was extracted every 2 h for Ni hydroxide and NiCo hydroxide and every 10 h for Co hydroxide. The composition of electrolyte after glycerol oxidation was identified based on the retention times of HPLC elution peaks of the individual standard sample. The product concentration was determined by the calibration curves of standard solutions with given concentrations. Detailed information about reaction product analysis can be found in Supplementary Note 3.

**DFT calculation**. Spin-polarized DFT simulations were performed using Vienna ab initio Simulation package (VASP, version 5.4.4)[59,60]. The projector-augmented wave (PAW) potential[61] and the Perdew–Burke–Ernzerhof (PBE) functional based generalized gradient approximation (GGA)[62] were employed. The DFT-D3 method with Becke–Jonson damping was adopted to correct the weak van der Waals interactions for layer structures[63,64]. The calculated density of state (DOS) was employed to perform the calculation of Crystal Orbital Hamiltonian Population (COHP) by LOBSTER program[65]. More detailed information about the DFT calculations can be found in Supplementary Notes 12 and 13.

## Data availability

The data supporting the findings of this study are available within the article and its Supplementary Information. Supplementary dataset for the coordinates of all

structure using in DFT calculations is provided with this paper. The data that support the findings of this study are available from https://figshare.com/articles/dataset/GOR-Source_data_zip/19783591. Source data are provided with this paper.

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

## Acknowledgements
This work was supported by the National Natural Science Foundation of China (11975102, Y.C.); the State Key Laboratory of Pulp and Paper Engineering (2022PY03, Y.C.); the Guangdong Pearl River Talent Program (2017GC010281, Y.C.). The synchrotron experiments were carried out at Beamline 02B of the Shanghai Synchrotron Radiation Facility, which is supported by ME2 project under contract from National Natural Science Foundation of China (11227902, N.Z.). This work was also supported by the National Research Foundation of Korea (NRF) grant funded by the Korean government (MSIP) (NRF-2020M1A2A2080650 and NRF-2021R1A2C3004019, J.W.H.).

## Author contributions
Z.H. and Y.C. conducted the experiments. J.H. and J.W.H. are responsible for the DFT calculations. Z.H., Z.G., M.Z., N.Z., and Y.C. are responsible for the XAS measurement and analysis. Z.H., X.K., and Y.C. are responsible for the in-situ infrared spectra measurement and analysis. Z.H., J.H., J.W.H., and Y.C. designed the project, analyzed the results, and wrote the manuscript.

## Competing interests
The authors declare no competing interests.
