## [Peer Review File · Nature Communications]

REVIEWER COMMENTS

Reviewer #1 (Remarks to the Author):

The paper "Promoting biomass electrooxidation via ..." by He et al. describes a combined computational and experimental investigation of the electrooxidation of glycerol over NiCo hydroxide catalysts. Biomass electrooxidation is a potentially very promising route for production of transportation fuels and value-added chemicals. Also, the formic acid selectivity of 94% is good at the applied voltage and current density. Doped Ni and Co hydroxide catalysts have been studied for a while but the glycerol oxidation over the catalysts remains very interesting. In the following, I focus my review on the DFT calculations. Here, I noticed the following issues:

- 1) The calculations are not clearly described. Are the numbers in Figure 2b energies or Gibbs free energies. The text in line 167 makes me believe that these are Gibbs free energies but if this is the case, then the authors need to specify how they computed the Gibbs free energies and how they considered various environmental effects. Are the energies with reference to glycerol at a fugacity of 1bar or at a specific concentration (which one)? How have solvation effects be considered? If they have not been considered than this should be stated. How has the pH and the applied potential been considered in the free energy calculations?
- 2) All coordinates should be provided of all structures with their corresponding energies such that the calculations are more reproducible.
- 3) Are the surface models metallic or do they display a significant band gap? If they display a significant bandgap, did the authors consider the possibility of a charged state being more stable than the neutral state. This can particularly become relevant when oxygen atoms are being removed/added for complex oxides. The authors need to at least clearly describe their calculations. Here, I would also like to see at least a reference that the energies are converged with energy cut-off of 500 eV which sounds small for a surface model of such a complex oxide.
- 4) Have transition states been considered? How have they been determined?
- 5) The free energy diagram in figure 2 involves very large changes in energy which makes me wonder whether a catalytic cycle can actually been closed on the surface. The surface contains elementary steps that involve processes that are more than 4 eV exergonic (see Table S2). How is this possible? Doesn't this point to a very unstable surface that is practically not present under reaction conditions? Have the authors made initially a careful constraint thermodynamic analysis to ensure that they are studying a physically relevant surface model?

Reviewer #2 (Remarks to the Author):

This paper reports novel scientific and mechanistic insights on electrochemically synthesized NiCo hydroxide catalysts for glycerol electrooxidation reaction (GOR). Glycerol is generated as a valuable byproduct in the production of biodiesel. How to efficiently convert the glycerol to renewable fuels and chemicals is an ongoing and important project in the research communities because global glycerol production has currently exceeded the demand for glycerol. The GOR process is a promising technology that can produce valuable chemicals such as dihydroxyacetone, lactic acid, glycolic acid and formic acid by glycerol oxidation reaction at anode and hydrogen by hydrogen evolution reaction at cathode. The GOR have been mainly reported with precious metal-based catalysts such as Au, Pt and Pd due to their excellent electrocatalytic activity and stability. However, only a few works have focused on developing non-precious metal based catalysts using Ni, Co, and Mn. This work demonstrates that doping Co into Ni-hydroxide promotes the deintercalation of proton and oxygen anion from the catalyst surface and the oxygen vacancies formed in NiCo hydroxide during GOR facilitates the charge transfer from catalyst surface to cleaved molecules during the 2nd C-C bond cleavage, thereby leading to enhanced electrocatalytic GOR performances. The manuscript is well organized with both computational and experimental methods and the conclusions are written based on scientific evidences. In particular, efforts to understand the fundamental and mechanistic phenomenon happened on NiCo hydroxide catalyst surface for the GOR are interesting and convincing. Thus I recommend its acceptance for publication in Nature Communications after some minor revisions. Some specific comments are listed below;

1. Please provide more detailed experimental information in the manuscript or supporting information. For example, which detector (UV or RI) and LC column in HPLC equipment did the authors use for the product analysis? What are the used equations to calculate the glycerol conversion and product selectivity? This information can be helpful to understand the results for the readers.
2. Have the authors calculated the product yield and selectivity based on carbon (or mass) balance? The authors need to check the carbon balance values due to the possibility of formation of gas products such as CO or CO₂ from formic acid. I assume the Ni based hydroxide catalysts may generate lots of gaseous products because of its excellent C-C cleavage ability as shown in this study.
3. In Figures 1 and S13, the authors mentioned NiCo hydroxide exhibited the optimal GOR performance among all samples after ECSA normalization. In order to know and understand about why the Ni and Co combination is important and how the Co doping into Ni-hydroxide contributes to enhanced catalytic performance, I think the authors need to calculate and present more kinetic data such as mass activity and specific activity (based on generated GOR current) & reaction rate and TOF (based on glycerol conversion from LC analysis).
4. In Figure S33, the authors have optimized different bulk structures of (oxy)hydroxide models and lattice parameters to suggest the possible bulk structures with different hydrogen arrangement. Since the spin-polarized simulations were utilized to obtain the total energies of bulk structures with different

hydrogen arrangement, it is questionable how the authors have considered the initial spin configurations. According to the work by Tkalych et al. (J. Phys. Chem. C 2015, 119, 24315-24322), the three different initial spin configurations of ferromagnetic, antiferromagnetic, and parallel spin were considered to describe the bulk structure, and thus resulting in the different energies with different initial spin configurations. Did the authors take account of these three initial spin configurations for the optimization of bulk structure? I recommend the authors to apply these initial spin configurations to confirm whether the suggested bulk structure is still valid or not.

5. According to the Figure 2a, S18, and S30, the second C-C cleavage step on NiCo (oxy)hydroxide was shown to occur on Co sites. Is there any evidence that second C-C cleavage preferentially occurs via Co sites on NiCo (oxy)hydroxide rather than Ni sites?

6. In Figure 4a, the authors have correlated the reaction energy of desorption step (7A to 7B) to the oxygen vacancy formation energy. For NiCo (oxy)hydroxide, there are two different oxygen sites, each of which might has the different oxygen vacancy formation energy. Is there any underlying assumption or reason why the authors chose a particular oxygen site to obtain the oxygen vacancy formation energy for NiCo (oxy)hydroxide?

Reviewer #3 (Remarks to the Author):

The article is appealing for a broad audience and address a hot topic in the literature, i.e., biomass oxidation coupled to H₂ generation. Besides, the work focuses in metals oxides free of noble metals, which are materials that have potential to be applied in electrolizers.

The most important contribution in this paper is connected with the fundamental study of the mechanism of the electrooxidation of glycerol on metal oxides. It is an area with many open questions and certainly this paper will serve of starting point for many other future contributions.

The paper is well-written, the materials are fully characterized, the measurements have been well-performed, and the mechanism studied in detail. However, my main concerns are related to the lack of information in some experimental sections and the lack of discussion about other mechanistic possibilities.

Below you will find some specific comments.

Comments to the authors

About the electrochemical measurements.

- It is quite confusing the fact that in several figures the authors plotted the results vs. RHE and in the text they used the Ag-AgCl scale (for example figure 1d). I suggest using the RHE scale in all the manuscript.

- It was nice to see that the authors estimated the ECSA to indeed probe that NiCo is more active. it would be useful you to include the CVs in the SI.

- Please, adjust the scale of figure S12.

-As you have many figures in figure 1 the description is rather poor (and this consideration applies to other figures also). I think that an expert in the field should understand almost everything what is going on just by looking at the figure and it is not the case. For example, how to understand something about the EIS results without knowing the electrochemical potential at which the measurement was performed?

- About EIS measurements. Which is the contribution of these measurements? In my opinion they are not useful as presented. They can be deleted or moved to the SI. Another option would be to do a more elaborated discussion like in a previous publication of the same group referenced here.

About the reaction products

- The information about HPLC analysis is poor. There is not enough data to reproduce the results. Column? Eluent? Temperature? Etc etc etc.

- I am glad that the authors included the chromatograms in the paper, which is something usually avoided. It would be useful to zoom in some of the figures, otherwise we see only the main product and glycerol.

- I am also suggesting to zoom in the chromatograms because I wonder if there is not another product in a relevant concentration. For instance, it called my attention that you detected glycerate as the only C3 product. Why are these catalysts selective to the oxidation of the primary carbon? If you oxidize the secondary carbon you will produce DHA, which will be quickly converted to lactate in alkaline media as showed in this contribution (<https://doi.org/10.1016/j.apcatb.2020.119369>). It would be also interesting you to compare the results with this recent paper <https://pubs.acs.org/doi/10.1021/acscatal.1c04150>.

- It is difficult to evaluate the FTIR in situ results. The figure is small and it is hard to have an idea of the wavenumber of the bands. These color maps are nice but not very useful in my opinion in this work. Anyway, the results are similar to those in this paper <https://doi.org/10.1016/j.jelechem.2021.115198>. You could compare the result and use some info to improve the discussion. Something that the authors have not paid attention to and that I think it is important is the presence (or absence) of the characteristic broad band due to the generation of carbonate at around 1400cm⁻¹. The absence of this band (which product can not be quantified by HPLC) is in line with the almost 100% of faradaic efficiency.

About the mechanism

First to all, I wanted to say that it will certainly be an important contribution in a field plenty of open questions. We have some information about the oxidation of alcohols on metal oxides coming mainly from the field of heterogeneous catalysis. I have seen for example the adsorption of methanol both through the C and O atom ([https://doi.org/10.1016/S0926-860X\(96\)00236-0](https://doi.org/10.1016/S0926-860X(96)00236-0)). Have the authors performed the calculations adsorbing glycerol through one (in the first step) of the three OH groups? If it is not the case, why? It deserves to be fully justified.

- The authors observe a correlation between the formation of the -OOH and the electrocatalytic behavior. It agrees with the results showed in the paper of the link presented before where the blanks were plotted together with the oxidations curves (<https://doi.org/10.1016/j.jelechem.2021.115198>). The -OOH formation indicates also the increase of the oxidation state of the metal in the metallic oxide and some papers claim, as I said before, that the O of the OH group of the alcohol attack the metallic center. In my opinion the authors should discuss this possibility and clarify why they have discarded this, in principle, suitable option.

- In general, I found the discussion very interesting independently of how accurate it is. The last result that I would like to see is the current vs. time curves obtained in the experiments showed in figure 1d. My concern is due to the fact that all the discussions in the mechanism always consider the differences in activity between Ni, Co and NiCo obtained through CV (figure 1a and many others). However, the authors do not collect the samples (and identify the products) during the CV using, for example, a sample collector (<https://pubs.acs.org/doi/10.1021/ac101058t>). The products were identified after hours of chronoamperometries, then the products (mainly formate) are connected in some way with the activities obtained in the chronoamperometry and not in the CV.

Following with the previous concern. I have doubts connected with the difference in activity of mainly Ni and NiCo. Looking at the results of figure 5, I wonder what happens if the current is divided for the capacitance. Is indeed NiCo more active than Ni?

Point-to-Point Responses to Referees' Comments and Suggestions

First of all, we truly appreciate the referees for their valuable comments and suggestions, which enormously improve the quality and clarity of this manuscript. All of the comments and suggestions have been taken into account in the revised manuscript as follows.

Response to Reviewer #1

COMMENTS TO AUTHOR:

The paper "Promoting biomass electrooxidation via ..." by He et al. describes a combined computational and experimental investigation of the electrooxidation of glycerol over NiCo hydroxide catalysts. Biomass electrooxidation is a potentially very promising route for production of transportation fuels and value-added chemicals. Also, the formic acid selectivity of 94% is good at the applied voltage and current density. Doped Ni and Co hydroxide catalysts have been studied for a while but the glycerol oxidation over the catalysts remains very interesting. In the following, I focus my review on the DFT calculations. Here, I noticed the following issues:

Response: We thank the reviewer for acknowledging the importance of our work. All the concerns raised from the reviewer have been addressed in detail as follows.

1) The calculations are not clearly described. Are the numbers in Figure 2b energies or Gibbs free energies. The text in line 167 makes me believe that these are Gibbs free energies

Response: We sincerely appreciate the reviewer for notifying us of typological errors. In the original manuscript, Figure 2b contained the relative energy to show the reaction energy of each step rather than the Gibbs free energy. In this revision, we have additionally calculated the Gibbs free energy to substitute reaction energy, and the reaction energy profile was moved to Supplementary Information as **Supplementary**

Fig. 30. To avoid any misunderstandings, we have clarified several statements in the manuscript.

Page 8, line 194 in manuscript, changed from: “The Gibbs free energy diagrams of GOR on Ni, NiCo, and Co hydroxides are shown in Fig. 2b and supplementary Table S2.” To “The Gibbs free energy diagrams of GOR on Ni, NiCo, and Co hydroxides are shown in Fig. 2b, Supplementary Table 2, and Supplementary note 6.”

Page 9, line 196 in manuscript, changed from: “The dehydrogenation process of the adsorbed reaction intermediates (1→2, 2→3, 4→5) are found to be spontaneous with negative reaction energy.” To “The dehydrogenation process of the adsorbed reaction intermediates (1→2, 2→3, 4→5) are found to be spontaneous with negative changes in Gibbs free energies.”

Page 28, line 1 in Supplementary Information, the caption of Supplementary Table 3 changed from: “The Gibbs free energy of each step in glycerol oxidation reaction.” To “The reaction energy of each step in glycerol oxidation reaction.”

Page 27, line 8 in Supplementary Information, Supplementary Table 2 added: “The Gibbs free energy of each step in glycerol oxidation reaction.”

Page 8, Figure 2b in manuscript, moved to supporting information as **Supplementary Fig. 30.**

Supplementary Fig. 30. Reaction energy calculations. The calculated reaction energy profiles of glycerol oxidation reaction on Ni, NiCo and Co hydroxide. The numerical data are shown in **Supplementary Table 3**.

but if this is the case, then the authors need to specify how they computed the Gibbs free energies and how they considered various environmental effects. Are the energies with reference to glycerol at a fugacity of 1 bar or at a specific concentration (which one)? How have solvation effects be considered? If they have not been considered than this should be stated. How has the pH and the applied potential been considered in the free energy calculations?

Response: As suggested by the reviewer, detailed information about the computational method used for determining Gibbs free energy and the environmental effects that were considered were described as the follows.

The Gibbs free energy (ΔG) of glycerol oxidation on Ni, NiCo, and Co hydroxides was calculated by correcting the obtained total energy with zero-point energy and entropy. The calculation of Gibbs free energy was modeled using computational equation (A-1) proposed by Nørskov et al. (*J. Phys. Chem. B* 2004, 108, 46, 17886–17892):

$$\Delta G = \Delta E + \Delta ZPE - T\Delta S + eU_{NHE} + \Delta G_{pH} \quad (A-1)$$

where ΔE is the total energy difference, ΔZPE is the difference of zero-point energy, ΔS is the change of entropy, and T is the environment temperature. The eU_{NHE} denotes the contribution of electrode potential versus normal hydrogen electrode (NHE) with elementary charge e and G_{pH} term denotes the contribution of pH. By introducing the U_{RHE} term, which is the potential versus reversible hydrogen electrode (RHE), the equation (A-1) can further be converted to equation (A-2), where pH contribution included in the U_{RHE} term reflects the pH dependence.

$$\Delta G = \Delta E + \Delta ZPE - T\Delta S + eU_{RHE} \quad (\text{A-2})$$

In our calculation, the calculation of Gibbs free energy was performed to compare the catalytic activity of Ni, NiCo, and Co hydroxides for the glycerol oxidation by using equation (A-2) with $T = 298\text{K}$ and $U_{RHE} = 0\text{V}$. Moreover, calculated Gibbs free energies were referenced to glycerol, glyceric acid, glycolic acid, and formic acid at 1 atm of fugacity. As our study is focused on comparing intrinsic activity of glycerol oxidation over Ni, NiCo, and Co hydroxides with identical surface structures, reaction intermediates and products, the solvation effect was neglected from Gibbs free energy calculation to avoid complexity. The Gibbs free energy agrees with the calculated reaction energy in terms of catalytic activity trend. The **Figure A1** and **Table A1** show the Gibbs free energy of each reaction step.

Figure A1. The calculated Gibbs free energy profiles of glycerol oxidation reaction on Ni, NiCo and Co hydroxide.

Table A1. The Gibbs free energy of each step in glycerol oxidation reaction.

Step	Reaction	Co (eV)	NiCo (eV)	Ni (eV)
0 → 1	Glycerol adsorption	- 0.37	- 0.04	- 0.46
1 → 2	Dehydrogenation (Glyceraldehyde)	- 2.92	- 2.56	- 2.00
2 → 3	Dehydrogenation + 1 st Lattice Oxygen Attack (Glycerate)	- 3.14	- 3.93	- 3.00
3 → A1	Hydrogenation and O _v formation (Glyceric acid)	+ 0.23	+ 1.34	+ 1.03
A1 → A2	Desorption (Glyceric acid)	+ 0.54	+ 1.14	+ 0.50
3 → 4	Change configuration	- 0.42	- 0.32	- 0.22
4 → 5	Dehydration + 2 nd Lattice Oxygen Attack	- 3.80	- 2.25	- 2.37
5 → 6	1 st C-C cleavage	+ 1.32	+ 1.21	+ 1.36
6 → 7A	Hydrogenation	+ 1.69	+ 0.74	+ 0.54

7A → 7B	Desorption (Formic acid)	- 0.16	+ 0.39	+ 0.54
7B → B1	Hydrogenation (Glycolic acid)	+ 0.21	+ 0.65	+ 0.66
B1 → B2	Desorption (Glycolic acid)	- 0.23	+ 0.01	- 0.09
7B → 8	3 rd Lattice Oxygen Attack	- 2.38	- 1.91	- 2.07
8 → 9A	2 nd C-C cleavage	+ 2.02	+ 0.37	+ 1.58
9A → 9B	Hydrogenation	+ 0.19	+ 0.21	+ 0.09
9B → FS	Desorption (Formic acid)	- 0.39	- 0.09	+ 0.10

Figure A1 has been substituted to Figure 2b in manuscript.

Table A1 has been added in Supplementary Information as Supplementary Table 2.

Figure 4 a and e in manuscript were changed to:

Figure 4 a, The free energy barrier of the desorption step (7A→7B) and the energy for the deintercalation of one oxygen anion from the hydroxide lattice (E_{f_vac}) for Ni, NiCo, and Co hydroxide.

Figure 4 e, The **free energy barrier** of the 2nd C-C bond cleavage step (8→9A) and the *d*-band filling before 2nd C-C bond cleavage of Ni, NiCo, and Co hydroxide. NiCo_Ni and NiCo_Co represent the Ni sites and Co sites on NiCo hydroxide, respectively.

Page 9, line 219 in manuscript, added: “In addition, the reaction energy calculations were also performed (**Supplementary Fig. 30, and Supplementary Table 3**), which show consistent results with the calculated Gibbs free energy in terms of catalytic activity trend.”

Page 44, line 11 in Supplementary Information, added:

“**Supplementary note 6: Gibbs free energy calculation**

Detailed information about the computational method used for determining Gibbs free energy and the environmental effects that were considered were described as the follows.

The Gibbs free energy (ΔG) of glycerol oxidation on Ni, NiCo, and Co hydroxides was calculated by correcting the obtained total energy with zero-point energy and entropy. The calculation of Gibbs free energy was modeled using computational equation (28):

$$\Delta G = \Delta E + \Delta ZPE - T\Delta S + eU_{NHE} + \Delta G_{pH} \quad (28)$$

where ΔE is the total energy difference, ΔZPE is the difference of zero-point energy, ΔS is the change of entropy, and T is the environment temperature. The eU_{NHE} denotes the contribution of electrode potential versus normal hydrogen electrode (NHE) with elementary charge e and G_{pH} term denotes the contribution of pH. By introducing the U_{RHE} term, which is the potential versus reversible hydrogen electrode (RHE), the equation (28) can further be converted to equation (29), where pH contribution included

in the U_{RHE} term reflects the pH dependence.

$$\Delta G = \Delta E + \Delta ZPE - T\Delta S + eU_{RHE} \quad (29)$$

In our calculation, the calculation of Gibbs free energy was performed to compare the catalytic activity of Ni, NiCo, and Co hydroxides for the glycerol oxidation by using equation (29) with $T = 298\text{K}$ and $U_{RHE} = 0\text{V}$. Moreover, calculated Gibbs free energies were referenced to glycerol, glyceric acid, glycolic acid, and formic acid at 1 atm of fugacity. As our study is focused on comparing intrinsic activity of glycerol oxidation over Ni, NiCo, and Co hydroxides with identical surface structures, reaction intermediates and products, the solvation effect was neglected from Gibbs free energy calculation to avoid complexity. The Gibbs free energy agrees with the calculated reaction energy in terms of catalytic activity trend. The **Figure 2b** and **Supplementary Table 2** show the Gibbs free energy of each reaction step.”

2) All coordinates should be provided of all structures with their corresponding energies such that the calculations are more reproducible.

Response: We thank the reviewer for the valuable suggestion. We agree that providing coordinates for all the structures will aid readers in better understanding the structures in each reaction step and make the calculations to be more reproducible. Therefore, coordinates of all structures from the initial state to the final state were added as **Supplementary note 15** in Supplementary Information, page 56 to 251. Moreover, the corresponding total energy of each structure was also denoted on the same page.

Page 56 to 251 in Supplementary Information, added the coordinates of all structures from the initial state to the final state. (Due to space limit, we did not list all the information in this response letter. Please see the revised Supplementary Information for detailed information.)

3) Are the surface models metallic or do they display a significant band gap? If they display a significant bandgap, did the authors consider the possibility of a charged state being more stable than the neutral state. This can particularly become relevant when oxygen atoms are being removed/added for complex oxides. The authors need to at least clearly describe their calculations. Here, I would also like to see at least a reference that the energies are converged with energy cut-off of 500 eV which sounds small for a surface model of such a complex oxide.

Response: We thank the reviewer for the insightful comment. The reviewer expressed concern about the possibility of a charged state being more stable than the neutral state when the surface models exhibit a large band gap. We completely agree with the reviewer's comment. However, our calculation results indicate that the surface models are almost metallic with very small band gap, as shown in **Table A2**.

Table A2. The calculated band gap energies of bulk and surface models of Ni, NiCo, and Co hydroxide.

	Bulk model	Surface model
Cobalt	1.67 eV (direct band-gap)	Metallic
Nickel/Cobalt	2.66 eV (direct band-gap)	0.01 eV (direct band-gap)
Nickel	3.33 eV (direct band-gap)	0.61 eV (direct band-gap)

Zhang et al. (*Chem. Commun.*, 2020, 56, 15387) reported that the nanosheet model of CoOOH has very small band gap, while that of bulk model of CoOOH has much larger band gap. Zhu et al (*Angew. Chem. Int. Ed.* 2016, 55, 12465) also reported that nickel hydroxide nanosheets exhibit the metallic rather than the semi-conductor characteristics, resulting in higher performance in electrocatalytic urea oxidation. These phenomena are consistent with our calculation results, which indicate that surface models have a much smaller band gap than bulk models to show metallic properties. Therefore, we did not consider the possibility of a charged state being more stable than the neutral state since the surface models does not exhibit a significant band gap.

As suggested by reviewer, the manuscript has been updated to include band gap calculation results to provide more complete descriptions of our calculation.

Table A2 has been added in Supplementary Information as *Supplementary Table 6*.

Page 54, line 17 in Supplementary Information, changed from “*Supplementary note 8: Bulk structure optimization of oxyhydroxide*” to “*Supplementary note 14: Bulk and surface structures optimization of oxyhydroxide*”

Page 56, line 5 in Supplementary Information, added:

“The band gap calculations were conducted on both bulk and surface models. As shown in Supplementary Table 6, the band gap energies computed for Co, NiCo, and Ni hydroxides are 1.67, 2.66, and 3.33 eV, respectively, indicating semiconductor characteristics. However, the surface models exhibit almost metallic properties with band gap of 0.01 to 0.61 eV. Such results about the small band gap of hydroxide surface are consistent with the previous reports^{54, 55}.”

Relative references have been cited in the revised Supplementary Information:

Page 56, line 9 in Supplementary Information, “[54] Zhang, S., et al. The latest development of CoOOH two-dimensional materials used as OER catalysts. *Chem Commun (Camb)* 56, 15387-15405 (2020). [55] Zhu, X., et al. Metallic Nickel Hydroxide Nanosheets Give Superior Electrocatalytic Oxidation of Urea for Fuel Cells. *Angew Chem Int Ed Engl* 55, 12465-12469 (2016).” were cited in the revised Supplementary Information as reference [54] and [55] in “*Such results about the small band gap of hydroxide surface are consistent with the previous reports^{54, 55}.”*

In addition to the band gap calculations, the reviewer also suggested us to provide some references showing that the energies are converged with energy cut-off of 500 eV for the surface models. The following references have used 500 eV for surface models of metal (oxy)hydroxides:

[1] Mefford et al, Interpreting Tafel behavior of consecutive electrochemical reactions through combined thermodynamic and steady state microkinetic approaches, *Energy Environ. Sci.*, 2020, 13, 622.

[2] Zhang et al, Lattice oxygen activation enabled by high-valence metal sites for enhanced water oxidation, *Nat. Commun.* 11, 4066 (2020)

These references were cited in the revised Supplementary Information:

Page 53, line 15 in Supplementary Information, “[51] Mefford, J. T., Zhao, Z., Bajdich, M., Chueh, W. C. *Interpreting Tafel behavior of consecutive electrochemical reactions through combined thermodynamic and steady state microkinetic approaches. Energy & Environmental Science* 13, 622-634 (2020). [52] Zhang, N., et al. *Lattice oxygen activation enabled by high-valence metal sites for enhanced water oxidation. Nat. Commun.* 11, 4066 (2020).” were cited in the revised Supplementary Information as reference [51] and [52] in “*The surface models were established with a single layer of hydrogen-terminated (001) plane of 5 x 5 supercells, and energy cut-off of 500 eV⁵¹,⁵² with a 3 x 3 x 1 Γ -point-centered Monkhorst-Pack k-point mesh was employed.*”

4) Have transition states been considered? How have they been determined?

Response: We sincerely appreciate that the reviewer provided helpful comment to improve the quality of our manuscript. In our original manuscript, we only showed the reaction energy at each elementary step due to the computational efficiency, but according to the reviewer’s comment, we additionally performed the calculations for the identification of transition state and activation energy on 2nd C-C cleavage step, which is a rate determining step, using the climbing image nudged elastic band method (CI-NEB) in the revised manuscript (*The Journal of Chemical Physics*, 2000, 113(22): 9901-9904.). Based on the obtained activation energy on **Figure A2**, NiCo (oxy)hydroxide shows the lowest energy barrier with the activation energy of 1.70 eV while Co (oxy)hydroxide shows the highest energy barrier with 2.64 eV on 2nd C-C cleavage step. This result is consistent with the obtained free energy barriers that were previously calculated. The activation energy also shows strong correlation with the *d*-

band filling of metal site on hydroxide (**Figure A3**), similar to that of the free energy barrier shown in Fig. 4e in manuscript.

Figure A2. The calculation of activation energies of 2nd C-C cleavage step for Ni, NiCo and Co (oxy)hydroxides by transition state searching. **a**, The energy profile of 2nd C-C cleavage step including the corresponding activation energies. **b**, Images of initial states (IS), transition states (TS) and final states (FS) of Ni, NiCo and Co (oxy)hydroxides for 2nd C-C cleavage step.

Figure A3. The activation energy of the 2nd C-C bond cleavage step (8→9A) and the *d*-band filling before 2nd C-C bond cleavage of Ni, NiCo, and Co hydroxide. NiCo_Ni and NiCo_Co represent the Ni sites and Co sites on NiCo hydroxide, respectively.

Figure A2 has been added in Supplementary Information as **Supplementary Fig. 39**.

Figure A3 has been added in Supplementary Information as **Supplementary Fig. 40**.

Page 15, line 373 in manuscript, added:

“Similar correlation also found between activation energy and d-band filling (Supplementary Fig. 39-40, Supplementary note 8).”

Page 48, line 1 in Supplementary Information, added:

“Supplementary note 8: transition states on 2nd C-C cleavage step

*For 2nd C-C cleavage step, which is a rate determining step, the identification of transition state and the calculation of activation energy were performed by climbing image nudged elastic band method (CI-NEB)⁴¹. According to the obtained activation energy on **Supplementary Fig. 39**, NiCo (oxy)hydroxide shows the lowest energy barrier with activation energy of 1.70 eV while Co (oxy)hydroxide shows the highest energy barrier with 2.64 eV on 2nd C-C cleavage step. This result is consistent with the obtained reaction free energies. The activation energy also shows strong correlation with the d-band filling of metal site on hydroxide (**Supplementary Fig. 40**), similar to that of the free energy barrier shown in Fig. 4e of the manuscript.”*

The corresponding reference was cited in the revised Supplementary Information:

Page 48, line 4 in Supplementary Information, “[41] Henkelman G, U. B. P., Jónsson H. A climbing image nudged elastic band method for finding saddle points and minimum energy paths. *J. Chem. Phys.* 113, 9901-9904 (2000).” was cited in the revised Supplementary Information as reference [41] in “For 2nd C-C cleavage step, which is a rate determining step, the identification of transition state and the calculation of activation energy were performed by using the climbing image nudged elastic band method (CI-NEB)⁴¹.”

5) The free energy diagram in Figure 2 involves very large changes in energy which makes me wonder whether a catalytic cycle can actually be closed on the surface. The surface contains elementary steps that involve processes that are more than 4 eV exergonic (see Table S2). How is this possible? Doesn't this point to a very unstable surface that is practically not present under reaction conditions? Have the authors made initially a careful constraint thermodynamic analysis to ensure that they are studying a physically relevant surface model?

Response: we truly appreciate the reviewer’s helpful comments and informing us of unusual reaction energies which were more than 4 eV exergonic. We double checked all the total energies of each reaction step and noticed that there was a mathematical error calculating the reaction energy on step 2 to 3 while other reaction energies are remained same. We apologize for the mistake. In step 2 to 3, the corrected energies were -2.84, -3.01, and -3.70 eV for Ni, Co, and NiCo, respectively, and total energy of each step can be found in the revised Supplementary Information page 56 to 251 (coordinates section).

In addition to the correction of reaction energies for step 2 to 3, surface energy calculation has been performed to verify the physical reliability of our surface model using the following equation. The surface energy (γ) was calculated by equation of (A-3):

$$\gamma = \frac{(E_{\text{surface_total}} - 2E_{\text{bulk_total}})}{2A} \quad (\text{A-3}),$$

where $E_{\text{surface_total}}$ is the total energy of surface models with 16 metal, 32 oxygen and 16 hydrogen atoms. $E_{\text{bulk_total}}$ is the total energy of bulk models with 8 metal, 16 oxygen, and 8 hydrogen atoms while A indicates the surface area of slab model. As a result, the computed surface energies of hydrogen atoms with chainsaw-like placement of Co, NiCo, and Ni (oxy)hydroxide were 0.66, 0.53, and 0.58 J/m², respectively. Navrotsky et al. (*Science*, 2010, 330(6001): 199-201.) reported the surface energy of various hydrated metal oxides surfaces including hydrated Co₃O₄ (spinel) with 0.92±0.04 J/m². Moreover, Deng et al. (*Nature Communications*, 2017, 8(1): 15194.) calculated the surface energy of Co hydroxide on (0001) surface, which was found to be 22.532 meV/Å² or 0.36 J/m². No substantial differences in the computed surface energies of Co, NiCo, and Ni (oxy)hydroxides were observed when compared to the reported surface energies. Therefore, we believe that the surface with hydrogen atoms of chainsaw-like placement of Co, NiCo, and Ni (oxy)hydroxides does not exhibit the unphysical properties of surface models.

Page 54, line 2 in Supplementary Information, added:

“The surface energy (γ) was calculated by the equation (33).

$$\gamma = \frac{(E_{\text{surface_total}} - 2E_{\text{bulk_total}})}{2A} \quad (33)$$

where $E_{\text{surface_total}}$ is the total energy of surface models with 16 metal, 32 oxygen and 16 hydrogen atoms. $E_{\text{bulk_total}}$ is the total energy of bulk models with 8 metal, 16 oxygen, and 8 hydrogen atoms while A indicates the surface area of slab model.”

Page 55, line 18 in Supplementary Information, added:

“Moreover, the surface energy calculation was performed by using the equation (33). The computed surface energies of hydrogen atoms with chainsaw-like placement of Co, NiCo, and Ni (oxy)hydroxides were 0.66, 0.53, and 0.58 J/m², respectively.”

Fig. 2 b in manuscript, changed and moved to Supplementary Fig. 30

Supplementary Fig. 30. Reaction energy calculations. The calculated reaction energy profiles of glycerol oxidation reaction on Ni, NiCo and Co hydroxide. The numerical data are shown in **Supplementary Table 3**.

Supplementary Table 2 in Supplementary Information changed to: **Supplementary Table 3**. The reaction energy of each step in glycerol oxidation reaction.

Step	Reaction	Co (eV)	NiCo (eV)	Ni (eV)
0 → 1	Glycerol adsorption	- 1.08	- 0.90	- 1.24

1 → 2	Dehydrogenation (Glyceraldehyde)	- 2.30	- 1.80	- 1.36
2 → 3	Dehydrogenation + 1 st Lattice Oxygen Attack (Glycerate)	- 3.01	- 3.70	- 2.84
3 → A1	Hydrogenation and O _v formation (Glyceric acid)	+ 0.02	+ 1.00	+ 0.81
A1 → A2	Desorption (Glyceric acid)	+ 1.21	+ 1.85	+ 1.20
3 → 4	Change configuration	- 0.48	- 0.44	- 0.18
4 → 5	Dehydration + 2 nd Lattice Oxygen Attack	- 3.50	- 2.04	- 2.20
5 → 6	1 st C-C cleavage	+ 1.50	+ 1.52	+ 1.52
6 → 7A	Hydrogenation	+ 1.25	+ 0.29	+ 0.21
7A → 7B	Desorption (Formic acid)	+ 0.61	+ 1.09	+ 1.24
7B → B1	Hydrogenation (Glycolic acid)	- 0.06	+ 0.27	+ 0.28
B1 → B2	Desorption (Glycolic acid)	+ 0.35	+ 0.71	+ 0.64
7B → 8	3 rd Lattice Oxygen Attack	- 2.59	- 2.10	- 2.28
8 → 9A	2 nd C-C cleavage	+ 2.22	+ 0.51	+ 1.84
9A → 9B	Hydrogenation	- 0.23	- 0.04	- 0.35
9B → FS	Desorption (Formic acid)	+ 0.97	+ 1.16	+ 1.46

Response to Reviewer #2

COMMENTS TO AUTHOR:

This paper reports novel scientific and mechanistic insights on electrochemically synthesized NiCo hydroxide catalysts for glycerol electrooxidation reaction (GOR). Glycerol is generated as a valuable byproduct in the production of biodiesel. How to efficiently convert the glycerol to renewable fuels and chemicals is an ongoing and important project in the research communities because global glycerol production has currently exceeded the demand for glycerol. The GOR process is a promising technology that can produce valuable chemicals such as dihydroxyacetone, lactic acid, glycolic acid and formic acid by glycerol oxidation reaction at anode and hydrogen by hydrogen evolution reaction at cathode. The GOR have been mainly reported with precious metal-based catalysts such as Au, Pt and Pd due to their excellent electrocatalytic activity and stability. However, only a few works have focused on developing non-precious metal based catalysts using Ni, Co, and Mn. This work demonstrates that doping Co into Ni-hydroxide promotes the deintercalation of proton and oxygen anion from the catalyst surface and the oxygen vacancies formed in NiCo hydroxide during GOR facilitates the charge transfer from catalyst surface to cleaved molecules during the 2nd C-C bond cleavage, thereby leading to enhanced electrocatalytic GOR performances. The manuscript is well organized with both computational and experimental methods and the conclusions are written based on scientific evidences. In particular, efforts to understand the fundamental and mechanistic phenomenon happened on NiCo hydroxide catalyst surface for the GOR are interesting and convincing. Thus I recommend its acceptance for publication in Nature Communications after some minor revisions. Some specific comments are listed below;

Response: We thank the reviewer for acknowledging the importance of our work and for the valuable comments. All the concerns raised from the reviewer have been addressed in detail as follows.

1. Please provide more detailed experimental information in the manuscript or supporting information. For example, which detector (UV or RI) and LC column in HPLC equipment did the authors use for the product analysis? What are the used equations to calculate the glycerol conversion and product selectivity? This information can be helpful to understand the results for the readers.

Response: We thank the reviewer for the valuable comments and suggestions. The glycerol oxidation products were determined by high-performance liquid chromatography (HPLC, Agilent 1260), which was equipped with a Xtimate Sugar-H column and a differential refractive index detector. Long-term electrolysis reactions of glycerol oxidation were carried out at a constant potential of 1.624 V (vs. RHE) by chronoamperometry in 50 mL electrolyte of 1 M KOH with 0.1 M glycerol. 1.0 mL electrolyte was extracted every 2 h for Ni hydroxide and NiCo hydroxide and every 10 h for Co hydroxide, and then was diluted with 1.0 mL 0.51 M H₂SO₄ solution to adjust the pH below 7.0. 20 μ L diluted solution was injected into the column. 0.27 mL H₂SO₄ diluted in 1000 mL H₂O was used as eluent with a constant flow rate of 0.6 mL/min. All the electrolysis experiments were carried out at room temperature and all the HPLC measurements were performed with a column temperature of 65 $^{\circ}$ C.

The composition of electrolyte after glycerol oxidation was identified based on the retention times of HPLC elution peaks of the individual standard sample. The product concentration was determined by the calibration curves of standard solutions with given concentrations. The glycerol conversion ($\eta_{glycerol}$), product yields ($Y_{glycerate}$, $Y_{glycolate}$, and $Y_{formate}$, respectively), and product selectivity ($S_{glycerate}$, $S_{glycolate}$, and $S_{formate}$, respectively) based on carbon balance were calculated by the following equations:

$$\eta_{glycerol} = \frac{C_{0,glycerol} - C_{glycerol}}{C_{0,glycerol}} \times 100\% \quad (\text{A-4})$$

$$Y_{glycerate} = \frac{C_{glycerate}}{C_{0,glycerol}} \times 100\% \quad (\text{A-5})$$

$$Y_{glycolate} = \frac{C_{glycolate} \times \frac{2}{3}}{C_{0,glycerol}} \times 100\% \quad (\text{A-6})$$

$$Y_{formate} = \frac{C_{formate} \times \frac{1}{3}}{C_{0,glycerol}} \times 100\% \quad (\text{A-7})$$

$$S_{glycerate} = \frac{C_{glycerate}}{C_{0,glycerol} - C_{glycerol}} \times 100\% \quad (\text{A-8})$$

$$S_{glycolate} = \frac{C_{glycolate} \times \frac{2}{3}}{C_{0,glycerol} - C_{glycerol}} \times 100\% \quad (\text{A-9})$$

$$S_{formate} = \frac{C_{formate} \times \frac{1}{3}}{C_{0,glycerol} - C_{glycerol}} \times 100\% \quad (\text{A-10})$$

where the $C_{0,glycerol}$ and $C_{glycerol}$ are the initial and final concentration of glycerol, respectively; $C_{glycerate}$, $C_{glycolate}$, and $C_{formate}$ are the final concentrations of glycerate, glycolate, and formate, respectively.

The Faradaic efficiency (FE) calculations of the glycerol oxidation production are based on the following half-reactions:

The corresponding Faradaic efficiencies toward glycerate, glycolate, and formate are calculated based on the following equations:

$$FE_{glycerate} = \frac{C_{glycerate} \times 4}{Q_{total}} \times V \times F \times 100\% \quad (\text{A-14})$$

$$FE_{glycolate} = \frac{C_{glycolate} \times \frac{2}{3} \times 5}{Q_{total}} \times V \times F \times 100\% \quad (\text{A-15})$$

$$FE_{formate} = \frac{C_{formate} \times \frac{1}{3} \times 8}{Q_{total}} \times V \times F \times 100\% \quad (\text{A-16})$$

where $C_{glycerate}$, $C_{glycolate}$, and $C_{formate}$ are the final concentrations of glycerate, glycolate, and formate, respectively; V is the volume of the electrolyte solution; F is the Faraday's constant (96485 C/mol); Q_{total} is the total charge passed.

Page 20, line 491 in manuscript, added: “Detailed information about reaction product analysis can be found in *Supplementary Note 3*.”

Page 39, line 4 in Supplementary Information, added:

“*Supplementary note 3: product analysis*”

The glycerol oxidation products were determined by high-performance liquid chromatography (HPLC, Agilent 1260), which was equipped with a Xtimate Sugar-H column and a differential refractive index detector. Long-term electrolysis reactions of glycerol oxidation were carried out at a constant potential of 1.624 V (vs. RHE) by chronoamperometry in 50 mL electrolyte of 1 M KOH with 0.1 M glycerol. 1.0 mL electrolyte was extracted every 2 h for Ni hydroxide and NiCo hydroxide and every 10 h for Co hydroxide, and then was diluted with 1.0 mL 0.51 M H₂SO₄ solution to adjust the pH below 7.0. 20 μL diluted solution was injected into the column. 0.27 mL H₂SO₄ diluted in 1000 mL H₂O was used as eluent with a constant flow rate of 0.6 mL/min. All the electrolysis experiments were carried out at room temperature and all the HPLC measurements were performed with a column temperature of 65 °C.

The composition of electrolyte after glycerol oxidation was identified based on the retention times of HPLC elution peaks of the individual standard sample. The product concentration was determined by the calibration curves of standard solutions with given concentrations. The glycerol conversion (η_{glycerol}), product yields ($Y_{\text{glycerate}}$, $Y_{\text{glycolate}}$, and Y_{formate} , respectively), and product selectivity ($S_{\text{glycerate}}$, $S_{\text{glycolate}}$, and S_{formate} , respectively) based on carbon balance were calculated by the following equations:

$$\eta_{\text{glycerol}} = \frac{C_{0,\text{glycerol}} - C_{\text{glycerol}}}{C_{0,\text{glycerol}}} \times 100\% \quad (15)$$

$$Y_{\text{glycerate}} = \frac{C_{\text{glycerate}}}{C_{0,\text{glycerol}}} \times 100\% \quad (16)$$

$$Y_{\text{glycolate}} = \frac{C_{\text{glycolate}} \times \frac{2}{3}}{C_{0,\text{glycerol}}} \times 100\% \quad (17)$$

$$Y_{\text{formate}} = \frac{C_{\text{formate}} \times \frac{1}{3}}{C_{0,\text{glycerol}}} \times 100\% \quad (18)$$

$$S_{\text{glycerate}} = \frac{C_{\text{glycerate}}}{C_{0,\text{glycerol}} - C_{\text{glycerol}}} \times 100\% \quad (19)$$

$$S_{\text{glycolate}} = \frac{C_{\text{glycolate}} \times \frac{2}{3}}{C_{0,\text{glycerol}} - C_{\text{glycerol}}} \times 100\% \quad (20)$$

$$S_{\text{formate}} = \frac{C_{\text{formate}} \times \frac{1}{3}}{C_{0,\text{glycerol}} - C_{\text{glycerol}}} \times 100\% \quad (21)$$

where the $C_{0,\text{glycerol}}$ and C_{glycerol} are the initial and final concentration of glycerol, respectively; $C_{\text{glycerate}}$, $C_{\text{glycolate}}$, and C_{formate} are the final concentrations of glycerate, glycolate, and formate, respectively.

The Faradaic efficiency calculations of the glycerol oxidation production are based on the following half-reactions:

The corresponding Faradaic efficiencies (FE) toward glycerate, glycolate, and formate are calculated based on the following equations:

$$FE_{\text{glycerate}} = \frac{C_{\text{glycerate}} \times 4}{Q_{\text{total}}} \times V \times F \times 100\% \quad (25)$$

$$FE_{\text{glycolate}} = \frac{C_{\text{glycolate}} \times \frac{2}{3} \times 5}{Q_{\text{total}}} \times V \times F \times 100\% \quad (26)$$

$$FE_{\text{formate}} = \frac{C_{\text{formate}} \times \frac{1}{3} \times 8}{Q_{\text{total}}} \times V \times F \times 100\% \quad (27)$$

where $C_{\text{glycerate}}$, $C_{\text{glycolate}}$, and C_{formate} are the final concentrations of glycerate, glycolate, and formate, respectively; V is the volume of the electrolyte solution; F is the Faraday's constant (96485 C/mol); Q_{total} is the total charge passed."

2. Have the authors calculated the product yield and selectivity based on carbon (or mass) balance? The authors need to check the carbon balance values due to the possibility of formation of gas products such as CO or CO₂ from formic acid. I assume

the Ni based hydroxide catalysts may generate lots of gaseous products because of its excellent C-C cleavage ability as shown in this study.

Response: We thank the reviewer for the valuable comments and suggestions. **Figure A4** shows the evolution of the concentrations of glycerol and reaction products as a function of electrolysis time, which was obtained from the HPLC measurement. For all samples, formate was detected as the main product, while glycerate and glycolate were detected as the minor products. The corresponding product yield and selectivity based on carbon balance are calculated by the equations (A-2 to A-7) and the calculated results are shown in **Figure A5** and **Figure A6**. For NiCo and Ni hydroxide, the total product yields were found to be close to the amount of converted glycerol, which suggested that all the products of GOR have been identified. Consistently, the total selectivity of Ni and NiCo hydroxide was found to be close to 100%. We believe that although the Ni-based hydroxide exhibited outstanding C-C cleavage ability, they did not over-oxidize the formic acid product to CO or CO₂ gaseous phases.

Figure A4. Concentration of glycerol and GOR products as a function of electrolysis time for different hydroxides. a-b Ni hydroxide, c-d NiCo hydroxide, e-f Co hydroxide.

Figure A5. The evolution of glycerol conversion and product yields as a function of electrolysis time at the potential of 1.624 V (vs. RHE) in 50 mL electrolyte of 1.0 M KOH with 0.1 M glycerol.

Figure A6. The selectivity of different products after electrolyzing at 1.624 V (vs. RHE) for 10 h in 50 mL electrolyte of 1.0 M KOH with 0.1 M glycerol.

Figure A4 has been added in Supplementary Information as **Supplementary Fig. 21**.

Page 6, line 140 in manuscript, added: *“For all samples, formate was detected as the main product, while glycerate and glycolate were detected as the minor products (Supplementary Fig. 21).”*

Page 6, line 152 in manuscript, changed from: *“As shown in the evolution of glycerol conversion and product yields as a function of electrolysis time (Fig. 1d), NiCo hydroxide exhibited the highest glycerol conversion rate and product yields rate.”*

To *“The glycerol conversion and product yields were calculated based on carbon balance (Supplementary note 3). As shown in Fig. 1d, NiCo hydroxide exhibited the highest glycerol conversion rate and product yields rate.”*

3. In Figures 1 and S13, the authors mentioned NiCo hydroxide exhibited the optimal GOR performance among all samples after ECSA normalization. In order to know and

understand about why the Ni and Co combination is important and how the Co doping into Ni-hydroxide contributes to enhanced catalytic performance, I think the authors need to calculate and present more kinetic data such as mass activity and specific activity (based on generated GOR current) & reaction rate and TOF (based on glycerol conversion from LC analysis).

Response: We thank the reviewer for the valuable suggestions. As suggested by the reviewer, we further calculate the reaction rate, turnover frequency (TOF) and mass activity to evaluate the intrinsic activity of the catalysts.

a). Reaction rate calculation:

The reaction rate for formate production was calculated based on the HPLC results. **Figure A7** shows the evolution of formate production rate as a function of electrolysis time. The reason why formate production rate decreases with the increase of electrolysis time is the depletion of glycerol in the electrolyte. The formate production rate exhibited a trend of NiCo > Ni > Co hydroxide, indicating the fastest GOR kinetic of NiCo hydroxide. This result is consistent with the what we observed in electrochemical measurement.

Figure A7. Formate production rate of Ni, NiCo and Co hydroxide. The evolution of formate production rate as a function of electrolysis time.

b). TOF calculation:

The TOF of formate production on Ni, NiCo and Co hydroxide was calculated by the following equation:

$$TOF (formate) = \frac{j}{\alpha n F} \times FE_{formate} \quad (A-17)$$

where j is the current density at a given potential obtained from electrochemical measurement ; α is the number of electron transfer to form one formate molecule; N is the number of active sites; F is the Faraday constant, equaling to 96485 C/mol; $FE_{formate}$ is the Faradaic efficiency toward formate. According to the equation (A-16), the $FE_{formate}$ for Ni, NiCo, and Co hydroxide were calculated to be 100%, 100%, and 57.7%, respectively.

The number of active sites per unit area was estimated by the follow equation:

$$N = \left(\frac{\text{number of active atom/unit cell}}{\text{volumn of unit cell}} \right)^{\frac{2}{3}} \times A_{ECSA} \quad (A-18)$$

where A_{ECSA} is the electrochemical surface area, which estimated by the double-layer capacitance (C_{dl}):

$$A_{ECSA}(NiOOH) = \frac{3.100 \text{ mF cm}^{-2}}{40 \mu\text{F cm}^{-2} \text{ per cm}^2_{ECSA}} = 77.5 \text{ cm}^2_{ECSA} \quad (A-19)$$

$$A_{ECSA}(NiCoOOH) = \frac{4.475 \text{ mF cm}^{-2}}{40 \mu\text{F cm}^{-2} \text{ per cm}^2_{ECSA}} = 111.875 \text{ cm}^2_{ECSA} \quad (A-20)$$

$$A_{ECSA}(CoOOH) = \frac{0.910 \text{ mF cm}^{-2}}{40 \mu\text{F cm}^{-2} \text{ per cm}^2_{ECSA}} = 22.75 \text{ cm}^2_{ECSA} \quad (A-21)$$

In our work, the computed values of optimized lattice parameters of (oxy)hydroxide models with chainsaw-like arrangement of hydrogen were as follow: CoOOH ($a = b = 6.091 \text{ \AA}$; $c = 9.124 \text{ \AA}$), NiCoOOH ($a = b = 5.948 \text{ \AA}$; $c = 9.022 \text{ \AA}$), and NiOOH ($a = b = 5.868 \text{ \AA}$; $c = 9.063 \text{ \AA}$), which contain 8 metal atoms, 16 oxygen atoms (half of them are exposed), and 8 hydrogen atoms. The exposed oxygen atoms act as the active sites for glycerol oxidation. Therefore, the number of active sites of Ni, NiCo, and Co (oxy)hydroxide were calculated to be:

$$\begin{aligned} N(NiOOH) &= \left(\frac{8 \text{ atoms/unit cell}}{5.868 \text{ \AA} \times 5.868 \text{ \AA} \times 9.063 \text{ \AA}} \right)^{\frac{2}{3}} \times 77.5 \text{ cm}^2_{ECSA} \\ &= 6.74 \times 10^{16} \end{aligned} \quad (A-22)$$

$$N(\text{NiCoOOH}) = \left(\frac{8 \text{ atoms/unit cell}}{5.948 \text{ \AA} \times 5.948 \text{ \AA} \times 9.022 \text{ \AA}} \right)^{\frac{2}{3}} \times 111.875 \text{ cm}_{\text{ECSA}}^2 \quad (\text{A-23})$$

$$= 9.58 \times 10^{16}$$

$$N(\text{CoOOH}) = \left(\frac{8 \text{ atoms/unit cell}}{6.091 \text{ \AA} \times 6.091 \text{ \AA} \times 9.124 \text{ \AA}} \right)^{\frac{2}{3}} \times 22.75 \text{ cm}_{\text{ECSA}}^2 \quad (\text{A-24})$$

$$= 1.87 \times 10^{16}$$

Therefore, the formula for calculating TOF can be written in the following format:

$$\text{TOF}(\text{NiOOH}) = \frac{j(A \text{ cm}^{-2}) \times A(\text{cm}^2) \times 6.02 \times 10^{23}(\text{mol}^{-1}) \times 100\%}{\frac{8}{3} \times 6.74 \times 10^{16} \times 96485 (A \text{ s mol}^{-1})} \quad (\text{A-25})$$

$$\text{TOF}(\text{NiCoOOH}) = \frac{j(A \text{ cm}^{-2}) \times A(\text{cm}^2) \times 6.02 \times 10^{23}(\text{mol}^{-1}) \times 100\%}{\frac{8}{3} \times 9.58 \times 10^{16} \times 96485(A \text{ s mol}^{-1})} \quad (\text{A-26})$$

$$\text{TOF}(\text{CoOOH}) = \frac{j(A \text{ cm}^{-2}) \times A(\text{cm}^2) \times 6.02 \times 10^{23}(\text{mol}^{-1}) \times 57.7\%}{\frac{8}{3} \times 1.87 \times 10^{16} \times 96485(A \text{ s mol}^{-1})} \quad (\text{A-27})$$

Specifically, the TOF of all samples at 1.4 V (vs. RHE) can be calculated to be:

$$\text{TOF}(\text{NiOOH}) = \frac{1.031 \times 10^{-1} \times 6.02 \times 10^{23} \times 100\%}{\frac{8}{3} \times 6.74 \times 10^{16} \times 96485} = 3.579 \text{ s}^{-1} \quad (\text{A-28})$$

$$\text{TOF}(\text{NiCoOOH}) = \frac{1.981 \times 10^{-1} \times 6.02 \times 10^{23} \times 100\%}{\frac{8}{3} \times 9.58 \times 10^{16} \times 96485} = 4.838 \text{ s}^{-1} \quad (\text{A-29})$$

$$\text{TOF}(\text{CoOOH}) = \frac{1.606 \times 10^{-3} \times 6.02 \times 10^{23} \times 57.7\%}{\frac{8}{3} \times 1.87 \times 10^{16} \times 96485} = 0.116 \text{ s}^{-1} \quad (\text{A-30})$$

Figure A8 shows the TOF values for GOR on Ni, NiCo, and Ni hydroxide at a potential of 1.4 V (vs. RHE). NiCo hydroxide exhibits a TOF value of 4.838 s⁻¹, which is much higher than that of Ni hydroxide (3.579 s⁻¹) and Co hydroxide (0.119 s⁻¹). This result suggests the highest intrinsic GOR activity of NiCo hydroxide.

Figure A8. The turnover frequency (TOF) calculation results. TOF for GOR on Ni, NiCo, and Ni hydroxide at a potential of 1.4 V (vs. RHE).

c). Mass activity calculation:

The mass activities of Ni and NiCo hydroxide were evaluated by normalizing the current density by loading mass obtained from inductively coupled plasma-optical emission spectrometry (ICP-OES) measurement. As shown in **Figure A9**, NiCo hydroxide exhibited a higher mass activity than Ni hydroxide. This result is consistent with the one we obtain by using ECSA for activity normalization (Supplementary Fig. 15 in the Supplementary Information), both suggesting that Co doping can effectively enhance the intrinsic activity of Ni hydroxide for GOR.

Figure A9. Mass activity of Ni and NiCo hydroxide. The current density was normalized by loading mass obtained from inductively coupled plasma-optical emission spectrometry (ICP-OES) measurement.

Figure A7 has been added in Supplementary Information as **Supplementary Fig. 23**.

Figure A8 has been added in Supplementary Information as **Supplementary Fig. 16**.

Figure A9 has been added in Supplementary Information as **Supplementary Fig. 17**.

Page 4, line 114 in manuscript, added: “*In addition, NiCo hydroxide exhibited a higher mass activity than Ni hydroxide (Supplementary Fig. 17), suggesting that Co doping can effectively enhance the intrinsic activity of Ni hydroxide for GOR.*”

Page 6, line 149 in manuscript, added: “*The formate production rate exhibited a trend of NiCo > Ni > Co hydroxide (Supplementary Fig. 23), indicating the fastest GOR kinetic of NiCo hydroxide. This result is consistent with the one we observed in electrochemical measurement (Fig. 1a-c).*”

Page 4, line 113 in manuscript, added: “*Turnover frequency results provide the same conclusion (Supplementary note 2, Supplementary Fig. 16).*”

Page 37, line 19 in Supplementary Information, added:

“**Supplementary note 2: turnover frequency (TOF) calculation**

The TOF of formate production on Ni, NiCo and Co hydroxide was calculated by the following equation:

$$TOF = \frac{j}{\alpha NF} \times FE_{formate} \quad (1)$$

where j is the current density at a given potential obtained from electrochemical measurement; α is the number of electron transfer to form one formate molecule; N is the number of active sites; F is the Faraday constant, equaling to 96485 C/mol; $FE_{formate}$ is the Faradaic efficiency toward formate. According to the equation (27), the $FE_{formate}$ for Ni, NiCo, and Co hydroxide were calculated to be 100%, 100%, and 57.7%, respectively.

The number of active sites per unit area was estimated by the follow equation:

$$N = \left(\frac{\text{number of active atom/unit cell}}{\text{volumn of unit cell}} \right)^{\frac{2}{3}} \times A_{ECSA} \quad (2)$$

where A_{ECSA} is the electrochemical surface area, which estimated by the double-layer capacitance (C_{dl}):

$$A_{ECSA}(NiOOH) = \frac{3.100 \text{ mF cm}^{-2}}{40 \mu\text{F cm}^{-2} \text{ per cm}_{ECSA}^2} = 77.5 \text{ cm}_{ECSA}^2 \quad (3)$$

$$A_{ECSA}(NiCoOOH) = \frac{4.475 \text{ mF cm}^{-2}}{40 \mu\text{F cm}^{-2} \text{ per cm}_{ECSA}^2} = 111.875 \text{ cm}_{ECSA}^2 \quad (4)$$

$$A_{ECSA}(CoOOH) = \frac{0.910 \text{ mF cm}^{-2}}{40 \mu\text{F cm}^{-2} \text{ per cm}_{ECSA}^2} = 22.75 \text{ cm}_{ECSA}^2 \quad (5)$$

In our work, the computed values of optimized lattice parameters of (oxy)hydroxide models with chainsaw-like arrangement of hydrogen were as follow: $CoOOH$ ($a = b = 6.091 \text{ \AA}$; $c = 9.124 \text{ \AA}$), $NiCoOOH$ ($a = b = 5.948 \text{ \AA}$; $c = 9.022 \text{ \AA}$), and $NiOOH$ ($a = b = 5.868 \text{ \AA}$; $c = 9.063 \text{ \AA}$), which contain 8 metal atoms, 16 oxygen atoms (half of them are exposed), and 8 hydrogen atoms. The exposed oxygen atoms act as the active sites for glycerol oxidation. Therefore, the number of active sites of Ni, NiCo, and Co (oxy)hydroxide were calculated to be:

$$\begin{aligned} N(NiOOH) &= \left(\frac{8 \text{ atoms/unit cell}}{5.868 \text{ \AA} \times 5.868 \text{ \AA} \times 9.063 \text{ \AA}} \right)^{\frac{2}{3}} \times 77.5 \text{ cm}_{ECSA}^2 \\ &= 6.74 \times 10^{16} \end{aligned} \quad (6)$$

$$\begin{aligned} N(NiCoOOH) &= \left(\frac{8 \text{ atoms/unit cell}}{5.948 \text{ \AA} \times 5.948 \text{ \AA} \times 9.022 \text{ \AA}} \right)^{\frac{2}{3}} \times 111.875 \text{ cm}_{ECSA}^2 \\ &= 9.58 \times 10^{16} \end{aligned} \quad (7)$$

$$\begin{aligned} N(CoOOH) &= \left(\frac{8 \text{ atoms/unit cell}}{6.091 \text{ \AA} \times 6.091 \text{ \AA} \times 9.124 \text{ \AA}} \right)^{\frac{2}{3}} \times 22.75 \text{ cm}_{ECSA}^2 \\ &= 1.87 \times 10^{16} \end{aligned} \quad (8)$$

Therefore, the formula for calculating TOF can be written in the following format:

$$TOF(NiOOH) = \frac{j(A \text{ cm}^{-2}) \times A(\text{cm}^2) \times 6.02 \times 10^{23}(\text{mol}^{-1}) \times 100\%}{\frac{8}{3} \times 6.74 \times 10^{16} \times 96485 (A \text{ s mol}^{-1})} \quad (9)$$

$$\begin{aligned} TOF(NiCoOOH) &= \frac{j(A \text{ cm}^{-2}) \times A(\text{cm}^2) \times 6.02 \times 10^{23}(\text{mol}^{-1}) \times 100\%}{\frac{8}{3} \times 9.58 \times 10^{16} \times 96485 (A \text{ s mol}^{-1})} \end{aligned} \quad (10)$$

$$TOF(CoOOH) = \frac{j(A\text{ cm}^{-2}) \times A(\text{cm}^2) \times 6.02 \times 10^{23}(\text{mol}^{-1}) \times 57.7\%}{\frac{8}{3} \times 1.87 \times 10^{16} \times 96485(A\text{ s mol}^{-1})} \quad (11)$$

Specifically, the TOF of all samples at 1.4 V (vs. RHE) can be calculated to be:

$$TOF(NiOOH) = \frac{1.031 \times 10^{-1} \times 6.02 \times 10^{23} \times 100\%}{\frac{8}{3} \times 6.74 \times 10^{16} \times 96485} = 3.579\text{ s}^{-1} \quad (12)$$

$$TOF(NiCoOOH) = \frac{1.981 \times 10^{-1} \times 6.02 \times 10^{23} \times 100\%}{\frac{8}{3} \times 9.58 \times 10^{16} \times 96485} = 4.838\text{ s}^{-1} \quad (13)$$

$$TOF(CoOOH) = \frac{1.606 \times 10^{-3} \times 6.02 \times 10^{23} \times 57.7\%}{\frac{8}{3} \times 1.87 \times 10^{16} \times 96485} = 0.116\text{ s}^{-1} \quad (14)''$$

4. In Figure S33, the authors have optimized different bulk structures of (oxy)hydroxide models and lattice parameters to suggest the possible bulk structures with different hydrogen arrangement. Since the spin-polarized simulations were utilized to obtain the total energies of bulk structures with different hydrogen arrangement, it is questionable how the authors have considered the initial spin configurations. According to the work by Tkalych et al. (J. Phys. Chem. C 2015, 119, 24315-24322), the three different initial spin configurations of ferromagnetic, antiferromagnetic, and parallel spin were considered to describe the bulk structure, and thus resulting in the different energies with different initial spin configurations. Did the authors take account of these three initial spin configurations for the optimization of bulk structure? I recommend the authors to apply these initial spin configurations to confirm whether the suggested bulk structure is still valid or not.

Response: We thank the reviewer for the insightful questions and valuable suggestions. As suggested by reviewer, we have conducted bulk calculations with different initial spin configurations for each structure. As a result, we concluded that structures with chain-saw like placement of hydrogen were still the most stable compared to other structures. The **Figure A10** shows the possible initial spin configuration of each bulk structure and **Table A3** shows the corresponding total energies.

Figure A10. Possible initial spin configurations of bulk structures: **a**, Hydrogen atoms located at each bottom layer of oxygen, **b**, location of hydrogen atoms with chain-saw-like placement, and **c**, hydrogen atoms located only at the middle layer of bulk structure.

Table A3. Total energy of bulk structure with different initial spin configurations.

	FM	AFM1	AFM2	PS
Co (oxy)hydroxide structure (a)	Optimized to (b) structure	- 181.43 eV	-	- 181.38 eV
NiCo (oxy)hydroxide structure (a)	Optimized to (b) structure	Optimized to (b) structure	Optimized to (b) structure	Optimized to (b) structure
Ni (oxy)hydroxide structure (a)	Optimized to (b) structure	Optimized to (b) structure	-	Optimized to (b) structure
Co (oxy)hydroxide structure (b)	- 182.06 eV	- 182.23 eV	-	- 181.73 eV
NiCo (oxy)hydroxide structure (b)	- 170.73 eV	- 170.88 eV	- 170.88 eV	- 171.16 eV
Ni (oxy)hydroxide structure (b)	- 159.54 eV	- 159.37 eV	-	- 159.39 eV
Co (oxy)hydroxide structure (c)	- 182.06 eV	Optimized to (b) structure	-	- 181.87 eV
NiCo (oxy)hydroxide structure (c)	Optimized to (b) structure	Optimized to (b) structure	Optimized to (b) structure	- 170.70 eV
Ni (oxy)hydroxide structure (c)	- 159.43 eV	- 159.41 eV	-	- 159.40 eV

Figure A10 has been added in Supplementary Information as **Supplementary Fig. 48**.

Table A3 has been added in Supplementary Information as **Supplementary Table 5**.

Supplementary Fig. 47 in Supplementary Information was changed to:

Supplementary Fig. 47. Configurations of bulk structures and the corresponding

total energies of Ni, NiCo, and Co hydroxide. a, Hydrogen atoms located at each

bottom layer of oxygen. b, location of hydrogen atoms with chain-saw-like

placement. c, hydrogen atoms located only at the middle layer of bulk structure.

Different bulk structures were composed of same number of atoms (metal: 8,

oxygen :16, hydrogen: 8).

Page 55, line 20 in Supplementary Information, added:

“We also conducted bulk calculations with different initial spin configurations for each structure. As a result (Supplementary Fig. 48 and Supplementary Table 5), we concluded that structures with chain-saw like placement of hydrogen were still the most stable compared to other structures.”

5. According to the Figure 2a, S18, and S30, the second C-C cleavage step on NiCo (oxy)hydroxide was shown to occur on Co sites. Is there any evidence that second C-C cleavage preferentially occurs via Co sites on NiCo (oxy)hydroxide rather than Ni sites?

Response: We thank the reviewer for the insightful question. We also considered alternative reaction pathways other than the one chosen in the manuscript for the second C-C cleavage step of NiCo (oxy)hydroxide. We compared the reaction energy of the second C-C cleavage step occurring on nickel sites to that of the preferred pathway involving Co sites. In addition to the second C-C cleavage step, we also considered the impact of reaction site for the hydrogenation step prior to the second C-C cleavage. As shown in **Figure A11a**, two different hydrogenation schemes were investigated. If the hydrogenation takes place from oxygen on Ni site, the glycolate would attack the Ni site oxygen to perform the second C-C cleavage while the opposite case allows second C-C cleavage of glycolate on Co sites. Our calculation results indicate that a step before the second C-C cleavage with Co site hydrogenation has 0.01 eV lower total energy compared to that of Ni site hydrogenation. Along with the aforementioned prediction, the reaction energy of second C-C cleavage on Ni sites reveals much higher energy (2.29 eV) compared to the case with second C-C cleavage on Co sites (0.51 eV). Therefore, we concluded that second C-C cleavage step on Co sites is energetically favorable than the Ni sites. The **Figure A11b** depicts the images of second C-C cleavage step that involves Ni sites and its corresponding reaction energy.

Figure A11. The reaction scheme of Ni site-based 2nd C-C cleavage step in NiCo (oxy)hydroxide and its corresponding reaction energy when 2nd C-C cleavage takes place in Ni site on NiCo (oxy)hydroxide. **a**, Possible hydrogenation schemes for the hydrogenation of glycolate. **b**, Structures of before and after 2nd C-C cleavage on Ni sites with the corresponding reaction energy.

Figure A11 has been added in Supplementary Information as **Supplementary Fig. 44. Page 16, line 388 in manuscript**, changed from “*These results suggest that the Co sites in NiCo hydroxide serve as the active sites for the 2nd C-C bond cleavage.*”

To “*These results suggest that the Co sites in NiCo hydroxide serve as the active sites for the 2nd C-C bond cleavage (Supplementary note 10).*”

Page 49, line 11 in Supplementary Information, added:

“Supplementary note 10: active site for 2nd C-C cleavage on NiCo hydroxide

*We also considered alternative reaction pathways other than the one chosen in the manuscript for the second C-C cleavage step of NiCo (oxy)hydroxide. We compared the reaction energy of the second C-C cleavage step occurring on nickel sites to that of the preferred pathway involving Co sites. In addition to the second C-C cleavage step, we also considered the impact of reaction site for the hydrogenation step prior to the second C-C cleavage. As shown in **Supplementary Fig. 44a**, two different hydrogenation schemes were investigated. If the hydrogenation takes place from oxygen on Ni site, the glycolate would attack the Ni site oxygen to perform the second C-C cleavage while the opposite case allows second C-C cleavage of glycolate on Co sites.*

*Our calculation results indicate that a step before the second C-C cleavage with Co site hydrogenation has 0.01 eV lower total energy compared to that of Ni site hydrogenation. Along with the aforementioned prediction, the reaction energy of second C-C cleavage on Ni sites reveals much higher energy (2.29 eV) compared to the case with second C-C cleavage on Co sites (0.51 eV). Therefore, we concluded that second C-C cleavage step on Co sites is energetically favorable than the Ni sites. The **Supplementary Fig. 44b** depicts the images of second C-C cleavage step that involves Ni sites and its corresponding reaction energy.”*

6. In Figure 4a, the authors have correlated the reaction energy of desorption step (7A to 7B) to the oxygen vacancy formation energy. For NiCo (oxy)hydroxide, there are two different oxygen sites, each of which might has the different oxygen vacancy formation energy. Is there any underlying assumption or reason why the authors chose a particular oxygen site to obtain the oxygen vacancy formation energy for NiCo (oxy)hydroxide?

Response: We thank the reviewer for the insightful question. The vacancy formation energy of two different oxygen sites (on Ni site or on Co site, denoted as $V_{\ddot{O}_{Ni}}$ or $V_{\ddot{O}_{Co}}$, respectively) have been calculated to be 1.44 eV and 1.05 eV, respectively, as shown in **Figure A12**. Oxygen atoms on Co sites in NiCo hydroxide were found to be more favorable to escape and form the vacancy than that on Ni sites, thus leading to a more obvious change in the valence state of Co in NiCo hydroxide as shown in the XAS results (Fig. 4d in manuscript).

Figure A12. The vacancy formation energy (E_{f_vac}) of two different oxygen sites ($V_{\ddot{O}_{Ni}}$ or $V_{\ddot{O}_{Co}}$) in NiCo hydroxide.

Correspondingly, the **Fig. 4a** in manuscript have been changed to:

Figure 4 a, The free energy barrier of the desorption step (7A→7B) and the energy for the deintercalation of one oxygen anion from the hydroxide lattice (E_{f_vac}) for Ni, NiCo, and Co hydroxide.

Response to Reviewer #3

COMMENTS TO AUTHOR:

The article is appealing for a broad audience and address a hot topic in the literature, i.e., biomass oxidation coupled to H₂ generation. Besides, the work focuses in metals oxides free of noble metals, which are materials that have potential to be applied in electrolizers.

The most important contribution in this paper is connected with the fundamental study of the mechanism of the electrooxidation of glycerol on metal oxides. It is an area with many open questions and certainly this paper will serve of starting point for many other future contributions.

The paper is well-written, the materials are fully characterized, the measurements have been well-performed, and the mechanism studied in detail. However, my main concerns are related to the lack of information in some experimental sections and the lack of discussion about other mechanistic possibilities. Below you will find some specific comments.

Response: We thank the reviewer for acknowledging the contribution of this work and for the valuable comments. All the concerns raised from the reviewer have been addressed in detail as follows.

About the electrochemical measurements.

- It is quite confusing the fact that in several figures the authors plotted the results vs. RHE and in the text they used the Ag-AgCl scale (for example figure 1d). I suggest using the RHE scale in all the manuscript.

Response: We thank the reviewer for the valuable suggestion. As suggested by the reviewer, all electrode potentials in the manuscript have been converted to be the ones versus reversible hydrogen electrode (RHE) according to the following equation:

$$E_{RHE} = E_{Ag/AgCl} + 0.059 \times pH + E_{Ag/AgCl}^{\theta} \quad (A-31)$$

where E_{RHE} is the converted potential versus RHE; $E_{Ag/AgCl}$ is the measured potential versus Ag/AgCl reference electrode; $E_{Ag/AgCl}^{\theta}$ is the standard potential of the Ag/AgCl reference electrode, equaling to 0.197 V.

Page 52, line 16 in Supplementary Information, added: “*All given potentials in the manuscript have been converted to be the ones versus reversible hydrogen electrode (RHE) according to the following equation:*

$$E_{RHE} = E_{Ag/AgCl} + 0.059 \times pH + E_{Ag/AgCl}^{\theta} \quad (32)$$

where E_{RHE} is the converted potential versus RHE; $E_{Ag/AgCl}$ is the measured potential versus Ag/AgCl reference electrode; $E_{Ag/AgCl}^{\theta}$ is the standard potential of the Ag/AgCl reference electrode, equaling to 0.197 V.”

Page 6, line 157 in manuscript, changed from “*The Faradaic efficiency of NiCo hydroxide for formate production was close to 100% when the applied potentials was higher than 0.45 V (vs. Ag/AgCl) (Fig. 1f, Supplementary Fig. 16)*”

To “*The Faradaic efficiency of NiCo hydroxide for formate production was close to 100% when the applied potentials was higher than 1.474 V (vs. RHE) (Fig. 1f, Supplementary Fig. 24)*”

Page 6, line 167 in manuscript, changed from “*NiCo hydroxide sample required only 0.15V (vs. Ag/AgCl) for formate formation, which is much lower than the 0.25 V for Ni hydroxide and 0.45 V for Co hydroxide, respectively.*”

To “*NiCo hydroxide sample required only 1.174 V (vs. RHE) for formate formation, which is much lower than the 1.274 V (vs. RHE) for Ni hydroxide and 1.474 V (vs. RHE) for Co hydroxide, respectively.*”

Page 11, line 241 in manuscript, changed from “*An oxidation current was observed when an anodic potential of 0.6 V (vs. Ag/AgCl) was first applied on NiCo hydroxide with 1.0 M KOH as the electrolyte.*”

To “*An oxidation current was observed when an anodic potential of 1.624 V (vs. RHE) was first applied on NiCo hydroxide with 1.0 M KOH as the electrolyte.*”

Page 11, line 246 in manuscript, changed from “Subsequently, a cathodic potential of -0.8 V (vs. Ag/AgCl) was applied on the catalyst and we observed no reduction current (Fig. 3a, solid line).”

To “Subsequently, a cathodic potential of 0.224 V (vs. RHE) was applied on the catalyst and we observed no reduction current (Fig. 3a, solid line).”

Page 12, line 292 in manuscript, changed from “Consistently, Co hydroxide showed Co-OOH Raman peak throughout whole electrochemical window ($0 - 0.4\text{ V}$ vs. Ag/AgCl) (Fig. 3e).”

To “Consistently, Co hydroxide showed Co-OOH Raman peak throughout whole electrochemical window ($1.024 - 1.424\text{ V}$ vs. RHE) (Fig. 3e).”

Page 52, line 5 in Supplementary Information, changed from “Electrochemical impedance spectra (EIS) were measured at the potential of 0.3 V vs. Ag/AgCl (overpotential of 150 mV) with an amplitude of 5 mV for GOR (HER).”

To “Electrochemical impedance spectra (EIS) were measured at the potential of 1.324 V (-0.15 V , vs. RHE) with an amplitude of 5 mV for GOR (HER).”

Page 52, line 8 in Supplementary Information, changed from “...which was obtained from the CV curves in the potential range of 0 to 0.1 V (vs. Ag/AgCl) with different rates from 100 to 500 mV/s .”

To “...which was obtained from the CV curves in the potential range of 1.024 to 1.124 V (vs. RHE) with different rates from 100 to 500 mV/s .”

Fig. 1 in manuscript changed to:

Fig. 1 | Evaluation of the electrocatalytic performance towards glycerol oxidation reaction (GOR). **a**, Cyclic voltammetry curves of Ni, NiCo, and Co hydroxide in 1 M KOH solution with (solid line) or without (dash line) 0.1 M glycerol. **b**, Tafel curves and **c**, electrochemical impedance spectroscopies of Ni, NiCo, and Co hydroxide for GOR. The EIS measurements were performed at the potential of 1.324 V (vs. RHE). **d**, The evolution of glycerol conversion and product yields as a function of electrolysis time at the potential of 1.624 V (vs. RHE) in 50 mL electrolyte of 1 M KOH with 0.1 M glycerol. The corresponding *i-t* curves were shown in **Supplementary Fig. S18**. **e**, The selectivity of different products after electrolyzing at 1.624 V (vs. RHE) for 10 h in 50 mL electrolyte of 1 M KOH with 0.1 M glycerol. **f**, The evolution of Faradaic efficiency as a function of applied potential on NiCo hydroxide with the same amount of total charge passed (1000 C). **g**, Chronopotentiometry curve for NiCo hydroxide acquired at the current density of 100 mA/cm². **h**, In-situ infrared spectra of Ni, NiCo and Co hydroxide obtained during the linear sweep voltammetry (LSV) measurement. The right panels are the enlarged figures of the dash line areas.

Fig. 3 in manuscript changed to:

Fig. 3 | The impact of proton deintercalation process on the dehydrogenation step during GOR. **a**, Multi-potential step curves of NiCo hydroxide. **b**, Deprotonation energy, i.e, energy required for the deintercalation of one proton from the lattice, of Ni, NiCo, and Co hydroxide. The green, blue, red, and white balls represent the Ni, Co, O, and H atoms, respectively. In-situ Raman spectroscopies for **c**, Ni hydroxide, **d**, NiCo hydroxide, and **e**, Co hydroxide in 1 M KOH without (left) and with (right) 0.1 M glycerol.

Fig. 4c,d in manuscript changed to:

Fig. 4 c, The Ni L-edge and **d**, Co L-edge soft X-ray absorption spectra (sXAS) of Ni, NiCo, and Co hydroxide after operating at 1.624 V (vs. RHE) in 1 M KOH with/without 0.1 M glycerol.

Supplementary Fig. 24 in Supplementary Information changed to:

Supplementary Fig. 24. The HPLC chromatograms of electrolyte after GOR using NiCo hydroxide as electrocatalyst at different applied potentials (1.324, 1.474, 1.624, 1.774 V vs. RHE) with the same amount of total charge passed (1000 C).

- It was nice to see that the authors estimated the ECSA to indeed probe that NiCo is more active. it would be useful you to include the CVs in the SI.

Response: We thank the reviewer for the valuable suggestion. **Figure A13** shows the corresponding CV curves with different scan rate (100 mV/s - 500 mV/s), which were used to estimate the double-layer capacitance of electrocatalysts.

Figure A13. Cyclic voltammetry curves with different scan rate (100 mV/s - 500 mV/s) used for electrochemical surface area evaluation. **a**, Ni hydroxide. **b**, NiCo hydroxide. **c**, Co hydroxide.

Figure A13 has been added in Supplementary Information as **Supplementary Fig. 14**. **Page 4, line 109 in manuscript**, changed: “*To rule out the impacts of microstructure, the current density was also normalized by electrochemical surface area (ECSA), which is estimated by double-layer capacitance (C_{dl}) (Supplementary Fig. 13).*”

To “*To rule out the impacts of microstructure, the current density was also normalized by electrochemical surface area (ECSA), which is estimated by double-layer capacitance (C_{dl}) (Supplementary Fig. 14 and Fig. 15).*”

- Please, adjust the scale of figure S12.

Response: We thank the reviewer for the valuable suggestion. The scale of **Supplementary Fig. 12** has been adjusted as shown in **Figure A14**.

Figure A14. Cyclic voltammetry curves (forward) of Co hydroxide in 1 M KOH solution with (solid line) or without (dash line) 0.1 M glycerol.

-As you have many figures in figure 1 the description is rather poor (and this consideration applies to other figures also). I think that an expert in the field should understand almost everything what is going on just by looking at the figure and it is not the case. For example, how to understand something about the EIS results without knowing the electrochemical potential at which the measurement was performed?

Response: We thank the reviewer for the valuable suggestion. We have double checked the description of the Figures in manuscript and added some necessary details.

Fig. 1 in manuscript changed to:

Fig. 1 | Evaluation of the electrocatalytic performance towards glycerol oxidation reaction (GOR). **a**, Cyclic voltammetry curves of Ni, NiCo, and Co hydroxide in 1 M KOH solution with (solid line) or without (dash line) 0.1 M glycerol. **b**, Tafel curves and **c**, electrochemical impedance spectroscopies of Ni, NiCo, and Co hydroxide for GOR. The EIS measurements were performed at the potential of 1.324 V (vs. RHE). **d**, The evolution of glycerol conversion and product yields as a function of electrolysis time at the potential of 1.624 V (vs. RHE) in 50 mL electrolyte of 1 M KOH with 0.1 M glycerol. The corresponding i-t curves were shown in **Supplementary Fig. 18**. **e**, The selectivity of different products after electrolyzing at 1.624 V (vs. RHE) for 10 h in 50 mL electrolyte of 1 M KOH with 0.1 M glycerol. **f**, The evolution of Faradaic efficiency as a function of applied potential on NiCo hydroxide with the same amount of total charge passed (1000 C). **g**, Chronopotentiometry curve for NiCo hydroxide acquired at the current density of 100 mA/cm². **h**, In-situ infrared spectra of Ni, NiCo and Co hydroxide obtained during the linear sweep voltammetry (LSV) measurement. The right panels are the enlarged figures of the dash line areas.

- About EIS measurements. Which is the contribution of these measurements? In my opinion they are not useful as presented. They can be deleted or moved to the SI. Another option would be to do a more elaborated discussion like in a previous publication of the same group referenced here.

Response: We thank the reviewer for the valuable suggestion. The EIS spectra of all samples exhibited a characteristic semicircle shape, which can be fitted by an Ohmic resistance (R_s), a constant phase element (CPE), and a charge-transfer resistance (R_{ct}) (**Figure A15**), in which the charge-transfer resistance is associate with the electrochemical reaction kinetics. NiCo hydroxide exhibited the smallest charge-transfer resistance value among all samples, indicating the fastest GOR kinetics.

Figure A15. The equivalent circuit used for electrochemical impedance spectroscopies fitting.

Figure A15 has been added in Supplementary Information as **Supplementary Fig. 13**. **Page 4, line 105 in manuscript**, added: “*The EIS spectra of all samples exhibited a characteristic semicircle shape, which were fitted by an Ohmic resistance (R_s), a constant phase element (CPE), and a charge-transfer resistance (R_{ct}) (Supplementary Fig. 13). NiCo hydroxide exhibited the smallest charge-transfer resistance value among all samples, indicating the fastest GOR kinetics.*”

About the reaction products

- The information about HPLC analysis is poor. There is not enough data to reproduce the results. Column? Eluent? Temperature? Etc etc etc.

Response: We thank the reviewer for the valuable suggestion. The glycerol oxidation products were determined by high-performance liquid chromatography (HPLC, Agilent 1260), which was equipped with a Xtimate Sugar-H column and a differential refractive index detector. Long-term electrolysis reactions of glycerol oxidation were carried out at a constant potential of 1.624 V (vs. RHE) by chronoamperometry in 50 mL electrolyte of 1 M KOH with 0.1 M glycerol. 1.0 mL electrolyte was extracted every 2 h for Ni hydroxide and NiCo hydroxide and every 10 h for Co hydroxide, and then was diluted with 1.0 mL 0.51 M H₂SO₄ solution to adjust the pH below 7.0. 20 μ L diluted solution was injected into the column. 0.27 mL H₂SO₄ diluted in 1000 mL H₂O was used as eluent with a constant flow rate of 0.6 mL/min. All the electrolysis experiments were carried out at room temperature and all the HPLC measurements were performed with a column temperature of 65 $^{\circ}$ C. As suggested by

the reviewer, we have added the detailed information of the HPLC analysis to the revised manuscript.

Page 39, line 5 in Supplementary Information, added:

“The glycerol oxidation products were determined by high-performance liquid chromatography (HPLC, Agilent 1260), which was equipped with a Xtimate Sugar-H column and a differential refractive index detector. Long-term electrolysis reactions of glycerol oxidation were carried out at a constant potential of 1.624 V (vs. RHE) by chronoamperometry in 50 mL electrolyte of 1 M KOH with 0.1 M glycerol. 1.0 mL electrolyte was extracted every 2 h for Ni hydroxide and NiCo hydroxide and every 10 h for Co hydroxide, and then was diluted with 1.0 mL 0.51 M H₂SO₄ solution to adjust the pH below 7.0. 20 μL diluted solution was injected into the column. 0.27 mL H₂SO₄ diluted in 1000 mL H₂O was used as eluent with a constant flow rate of 0.6 mL/min. All the electrolysis experiments were carried out at room temperature and all the HPLC measurements were performed with a column temperature of 65 °C.”

- I am glad that the authors included the chromatograms in the paper, which is something usually avoided. It would be useful to zoom in some of the figures, otherwise we see only the main product and glycerol.

Response: We thank the reviewer for the valuable suggestion. As suggested by the reviewer, the peaks of the minor products (glycerate and glycolate) in HPLC spectra have been magnified to be clearer as shown in **Figure A16** (right) and **Figure A17** (right).

Figure A16. The evolution of HPLC chromatograms as a function of

electrolyzing time. a, Ni hydroxide, **b**, Co hydroxide and **c**, NiCo hydroxide. The electrolysis experiments were carried out on a constant applied potential of 1.624 V (vs. RHE) in 50 mL electrolyte of 1 M KOH with 0.1 M glycerol. The right panels are the enlarged figures of the areas marked by the dash line.

Figure A17. The HPLC chromatograms of GOR products for NiCo hydroxide.

The electrolysis experiment was carried out at different applied potentials (1.324, 1.474, 1.624, 1.774 V vs. RHE) with the same amount of total charge passed (1000 C). The right panels are the enlarged figures of the dash line areas.

Figure A16 has been added in Supplementary Information as **Supplementary Fig. 20**.

Figure A17 has been added in Supplementary Information as **Supplementary Fig. 24**.

- I am also suggesting to zoom in the chromatograms because I wonder if there is not another product in a relevant concentration. For instance, it called my attention that you detected glycerate as the only C3 product. Why are these catalysts selective to the oxidation of the primary carbon? If you oxidize the secondary carbon you will produce DHA, which will be quickly converted to lactate in alkaline media as showed in this contribution (<https://doi.org/10.1016/j.apcatb.2020.119369>). It would be also interesting you to compare the results with this recent paper <https://pubs.acs.org/doi/10.1021/acscatal.1c04150>.

Response: We thank the reviewer for the insightful question and valuable comment. Lima et al. (*Applied Catalysis B: Environmental*, 2020, 279: 119369.) proposed the equivalent equilibria between glyceraldehyde and dihydroxyacetone (DHA) and

reported that DHA can convert to lactate in alkaline media by Cannizzaro reaction. We indeed observed the presence of lactate in the enlarged HPLC chromatograms (**Figure A18**), but its content is rather small compared to the main product (formate), which is negligible. In addition, the lattice oxygen attack of glyceraldehyde steps (2→3) on all samples were found to be with a negative reaction energy (-2.84 eV, -3.70 eV, and -3.01 eV for Ni, NiCo, and Co hydroxide, respectively) (**Figure A19**). This computational result suggested that glyceraldehyde is preferable to go through 1st lattice oxygen attack step to form glycerate, which is consistent with the experiment observation.

Figure A18. The HPLC chromatograms analysis of GOR products for NiCo hydroxide. The electrolysis experiment was carried out at a constant applied potential of 1.624 V (vs. RHE) for 10 h in 50 mL electrolyte of 1.0 M KOH with 0.1 M glycerol on NiCo hydroxide. The bottom panel is the enlarged figure.

Figure A19. Relative reaction energy of the lattice oxygen attack of glyceraldehyde step on Ni, NiCo, and Co hydroxide.

Morales et al. (*ACS Catalysis*, 2022, 12: 982-992.) reported that the composition of products of glycerol electrooxidation strongly depends on the applied potential, electrolyte concentration, and duration of electrolysis. They found the formic acid as the main products, which is consistent with our results. The most different point is that, Morales et al found the degradation of formate concentration after electrolyzing at 1.5 V (vs. RHE) for 4 h due to the further oxidation. In contrast, we found that the formate can remain stable even performing at 1.624 V (vs. RHE) for 10 h, which allow us to achieve a high yield of formate.

Figure A18 has been added into the Supplementary Information as **Supplementary Fig. 29**.

Page 42, line 10 in Supplementary Information, added: “*It is reported that there is an equilibrium between glyceraldehyde and dihydroxyacetone (DHA), which can convert to lactate in alkaline media by Cannizzaro reaction³⁷. In our work, the content of lactate in products is rather small compared to the main product (formate) (Supplementary Fig. 29), which is negligible. Therefore, we proposed the pathway that*

*glyceraldehyde undergoes the further dehydrogenation and the 1st lattice oxygen attack (2 → 3) to form glycerate intermediate (*OOC-CHOH-CH₂OH) for further oxidation.”*

The references mentioned by the reviewer have been cited in the revised Supplementary Information:

Page 42, line 10 in Supplementary Information, “[37] Lima, C. C., et al. *Highly active Ag/C nanoparticles containing ultra-low quantities of sub-surface Pt for the electrooxidation of glycerol in alkaline media. Applied Catalysis B: Environmental* 279, 119369 (2020).” was cited in the revised Supplementary Information as reference [37] in “It is reported that there is an equilibrium between glyceraldehyde and dihydroxyacetone (DHA), which can convert to lactate in alkaline media by Cannizzaro reaction³⁷.”

- It is difficult to evaluate the FTIR in situ results. The figure is small and it is hard to have an idea of the wavenumber of the bands. These color maps are nice but not very useful in my opinion in this work. Anyway, the results are similar to those in this paper <https://doi.org/10.1016/j.jelechem.2021.115198>. You could compare the result and use some info to improve the discussion. Something that the authors have not paid attention to and that I think it is important is the presence (or absence) of the characteristic broad band due to the generation of carbonate at around 1400cm⁻¹. The absence of this band (which product can not be quantified by HPLC) is in line with the almost 100% of faradaic efficiency.

Response: We thank the reviewer for the valuable comment and suggestion. As suggested by the reviewer, the in-situ FTIR spectra have been magnified to show the appearance of formate absorption bands clearer (**Figure A20**). As the applied potential increased, the absorption bands located at 1356, 1388, 1585 cm⁻¹ appeared on all three samples, which is corresponding to the C-O symmetric stretching, COO rocking, and C-O asymmetric stretching of formate, respectively (*Journal of Electroanalytical Chemistry*, 2021, 896: 115198.). NiCo hydroxide sample required only 1.174 V (vs.

RHE) for formate formation, which is much lower than the 1.274 V (vs. RHE) for Ni hydroxide and 1.474 V (vs. RHE) for Co hydroxide, respectively. However, the corresponding absorption bands of glycerate (1580 cm^{-1} and 1419 cm^{-1}) and glycolate (1580 cm^{-1} , 1410 cm^{-1} , 1326 cm^{-1} , and 1075 cm^{-1}) cannot be observed due to their low concentration (*Journal of Electroanalytical Chemistry*, 2021, 896: 115198.). In addition, the absence of the characteristic broad band of carbonate at around 1400 cm^{-1} suggested that formate can remain stable without further oxidation to form carbonate, which is consistent with the almost 100% of Faradic frequency for formate production.

Figure A20. a-c, In-situ infrared spectra of Ni (a), NiCo (b) and Co (c) hydroxide obtained during the linear sweep voltammetry (LSV) measurement. The right panels are the enlarged figures of the dash line areas.

Figure A20 has been added in manuscript as **Fig. 1h**.

Page 6, line 164 in manuscript, changed from: “As the applied potential increased, the absorption bands of formate located at 1356 , 1388 , 1585 cm^{-1} appeared on all three samples⁴⁷. NiCo hydroxide sample required only 0.15 V (vs. Ag/AgCl) for formate formation, which is much lower than the 0.25 V for Ni hydroxide and 0.45 V for Co hydroxide, respectively. This result is consistent with the electrochemical test and the products analysis, suggesting that NiCo hydroxide showed the highest activity towards the conversion of glycerol to formate.”

To “As the applied potential increased, the absorption bands located at 1356 , 1388 , 1585 cm^{-1} appeared on all three samples, which is corresponding to the C-O symmetric stretching, COO rocking, and C-O asymmetric stretching of formate, respectively^{47,48}. NiCo hydroxide sample required only 1.174 V (vs. RHE) for formate formation, which

is much lower than the 1.274 V (vs. RHE) for Ni hydroxide and 1.474 V (vs. RHE) for Co hydroxide, respectively. This result is consistent with the electrochemical test and the products analysis, suggesting that NiCo hydroxide showed the highest activity towards the conversion of glycerol to formate. However, the corresponding absorption bands of glycerate (1580 cm^{-1} and 1419 cm^{-1}) and glycolate (1580 cm^{-1} , 1410 cm^{-1} , 1326 cm^{-1} , and 1075 cm^{-1}) cannot be observed due to their low concentration. In addition, the absence of the characteristic broad band of carbonate at around 1400 cm^{-1} suggested that formate can remain stable without further oxidation to form carbonate, which is consistent with the almost 100% of Faradic frequency for formate production.”

The reference recommended by the reviewer and the related reference were added to the revised manuscript:

Page 6, line 166 in manuscript, “[47] Gomes, J. F., et al. Influence of silver on the glycerol electro-oxidation over AuAg/C catalysts in alkaline medium: A cyclic voltammetry and in situ FTIR spectroscopy study. *Electrochim. Acta* 144, 361-368 (2014). [48] Santiago, P. V. B., Lima, C. C., Bott-Neto, J. L., Fernández, P. S., Angelucci, C. A., Souza-Garcia, J. Perovskite oxides as electrocatalyst for glycerol oxidation. *J. Electroanal. Chem.* 896, 115198 (2021).” were cited in the revised manuscript as reference [47] and [48] in “which is corresponding to the C-O symmetric stretching, COO rocking, and C-O asymmetric stretching of formate, respectively^{47,48}.”

About the mechanism

First to all, I wanted to say that it will certainly be an important contribution in a field plenty of open questions. We have some information about the oxidation of alcohols on metal oxides coming mainly from the field of heterogeneous catal. I have seen for example the adsorption of methanol both through the C and O atom ([https://doi.org/10.1016/S0926-860X\(96\)00236-0](https://doi.org/10.1016/S0926-860X(96)00236-0)). Have the authors performed the calculations adsorbing glycerol through one (in the first step) of the three OH groups? If it is not the case, why? It deserves to be fully justified.

- The authors observe a correlation between the formation of the -OOH and the electrocatalytic behavior. It agrees with the results showed in the paper of the link presented before where the blanks were plotted together with the oxidations curves (<https://doi.org/10.1016/j.jelechem.2021.115198>). The -OOH formation indicates also the increase of the oxidation state of the metal in the metallic oxide and some papers claim, as I said before, that the O of the OH group of the alcohol attack the metallic center. In mi opinion the authors should discuss this possibility and clarify why they have discarded this, in principle, suitable option.

Response: We thank the reviewer for the valuable comment. We completely agree that it is important to justify the probability of metallic sites being attacked by glycerol's OH. To confirm that the proposed adsorption configuration of glycerol is more favorable than the attacking the metallic center, we have conducted the calculations of glycerol adsorption directly to the metallic site through OH for Ni, NiCo and Co hydroxides. As shown in **Figure A21**, the calculated results indicate that OH of glycerol is unstable in its attack on the metallic site of the surface. The unoptimized initial configurations that were modelled to attack the metallic site were spontaneously optimized to have configurations with OH to oxygen sites on the surface rather than the OH to the metal sites. This clearly shows that the glycerol is more favorable to be adsorbed on the surface through oxygen sites rather than the metal sites. As suggested by the reviewer, we have added the discussion regarding to the consideration of probability of metallic sites being attacked by OH in glycerol.

Figure A21. Adsorption structures of glycerol on metal site before and after optimization.

Figure A21 has been added in Supplementary Information as **Supplementary Fig. 25. Page 42, line 8 in Supplementary Information**, changed from “*These exposed lattice oxygen site then act as the active sites for glycerol adsorption (0 → 1).*”

To “*These exposed lattice oxygen site then act as the active sites for glycerol adsorption (0 → 1) (Supplementary note 5).*”

Page 43, line 21 in Supplementary Information, added:

“Supplementary note 5: active site of glycerol adsorption

To identify the active site of glycerol adsorption, we have conducted the calculations of the glycerol adsorption directly to the metallic site through OH for Ni, NiCo and Co hydroxides. As shown in Supplementary Fig. 25, the calculated results indicate that OH of glycerol is unstable in its attack on the metallic site of the surface. The unoptimized initial configurations that were modelled to attack the metallic site were spontaneously optimized to have configurations with OH to oxygen sites on the surface rather than the OH to the metal sites. This clearly shows that the glycerol is more favorable to be adsorbed on the surface through oxygen sites rather than the metal sites.”

- In general, I found the discussion very interesting independently of how accurate it is. The last result that I would like to see is the current vs. time curves obtained in the experiments showed in figure 1d. My concern is due to the fact that all the discussions in the mechanism always consider the differences in activity between Ni, Co and NiCo obtained through CV (figure 1a and many others). However, the authors do not collect the samples (and identify the products) during the CV using, for example, a sample collector (<https://pubs.acs.org/doi/10.1021/ac101058t>). The products were identified after hours of chronoamperometries, then the products (mainly formate) are connected in some way with the activities obtained in the chronoamperometry and not in the CV.

Response: We thank the reviewer for the valuable comment. **Figure A22** shows the evolution of current density as a function of electrolysis time in 1.0 M KOH with 0.1 M glycerol. NiCo hydroxide exhibited higher current density than Ni and Co hydroxide, suggesting its highest GOR activity, which is consistent with the CV measurement results. After 10 hours electrolysis, the current density of NiCo hydroxide become similar to that of Ni hydroxide due to the consumption of glycerol. The faster current density decay for NiCo hydroxide indicates the higher glycerol conversion, which is consistent with the HPLC results.

Figure A22. The evolution of current density as a function of electrolysis time 1.0 M KOH with 0.1 M glycerol.

Figure A22 has been added in Supplementary Information as **Supplementary Fig. 18**. **Page 5, line 128 in manuscript**, added: “*The corresponding i-t curves were shown in Supplementary Fig. 18.*”

As suggested by the reviewer, we collected the reaction products online during the LSV measurement and identify its composition by ex-situ HPLC measurement (**Figure A23**). The LSV measurement was carried out with a scan rate of 1 mV/s in a solution of 1.0 M KOH with 0.1 M glycerol. Aliquots of 0.3 mL electrolyte were collected with a flow rate of 3 $\mu\text{L/s}$. Thus, each collected sample represents the products of GOR generated in an interval of 100 mV. **Figure A23a** shows the LSV curve, and **Figure A23c** shows the evolution of the product concentrations as a function of the applied potential on NiCo hydroxide. As the applied potential increase, the concentration of formate shows a significant increase, while the concentration of glycerate and glycolate remain unchanged. This result suggests that formate is the main product, which is consistent with the HPLC results after long-time electrolysis.

Figure A23. Reaction product analysis during LSV measurements. a, LSV curve of NiCo hydroxide during online collection of reaction products for HPLC measurement. The scan rate is 1 mV/s. **b**, The ex-situ HPLC chromatograms of reaction products collecting at different potential during LSV measurement on NiCo hydroxide. **c**, The evolution of product concentration as a function of the applied potential, in which the product concentrations were obtained based on the corresponding HPLC chromatograms.

Figure A23 has been added in Supplementary Information as **Supplementary Fig. 22**.

Page 6 line 143 in manuscript, added: “*We also collected the reaction products online during the LSV measurement and identified its composition by ex-situ HPLC measurement (Supplementary Fig. 22). As the applied potential increase, the concentration of formate shows a significant increase, while the concentration of glycerate and glycolate remain unchanged. This result suggests that formate is the main product, which is consistent with the HPLC results after long-time electrolysis.*”

Page 40, line 9 in Supplementary Information, added: “*The reaction products were also collected online during the LSV measurement, and their compositions were quantified by ex-situ HPLC measurement^{35, 36}. The LSV measurement was carried out with a scan rate of 1 mV/s in a solution of 1.0 M KOH with 0.1 M glycerol. Aliquots of 0.3 mL electrolyte were collected with a flow rate of 3 μ L/s. Thus, each collected sample represents the products of GOR generated in an interval of 100 mV.*”

The reference recommended by the reviewer and the related reference were added to the revised manuscript:

Page 40, line 9 in Supplementary Information, “[35] Youngkook Kwon, M. T. M. K. Combining Voltammetry with HPLC: Application to Electro-Oxidation of Glycerol. *Anal. Chem.* 82, 5420–5424 (2010). [36] Santiago, P. V. B., Lima, C. C., Bott-Neto, J. L., Fernández, P. S., Angelucci, C. A., Souza-Garcia, J. Perovskite oxides as electrocatalyst for glycerol oxidation. *J. Electroanal. Chem.* 896, 115198 (2021).” were cited in the revised manuscript as reference [35] and [36] in “*We also collected the reaction products online during the LSV measurement and identify its composition by ex-situ HPLC measurement^{35, 36}.*”

Following with the previous concern. I have doubts connected with the difference in activity of mainly Ni and NiCo. Looking at the results of figure 5, I wonder what happen if the current is divided for the capacitance. Is indeed NiCo more active than Ni?

Response: We thank the reviewer for the valuable question. To evaluate the intrinsic activity of Ni, NiCo, and Co hydroxide for overall electrolysis coupling GOR

with hydrogen evolution reaction (HER), the current density was also normalized by the double-layer capacitance (C_{dl}). As shown in **Figure A24**, NiCo hydroxide achieved a current density of 11.96 A/F at the voltage of 1.5 V, which is much higher than that of Ni hydroxide (4.32 A/F) and Co hydroxide (1.20 A/F), suggesting that NiCo hydroxide indeed exhibited a much higher activity than the Ni hydroxide and Co hydroxide for overall electrolysis coupling GOR with HER.

Figure A24. Specific activity for overall electrolysis coupling HER and GOR.

LSV polarization curves normalized by double-layer capacitance for overall electrolysis coupling HER and GOR.

Figure A24 has been added in Supplementary Information as **Supplementary Fig. 46**. **Page 16, line 406 in manuscript**, added: “*After normalizing the current density by the double-layer capacitance (C_{dl}) (Supplementary Fig. 46), NiCo hydroxide achieved a current density of 11.96 A/F at the voltage of 1.5 V, which is much higher than that of Ni hydroxide (4.32 A/F) and Co hydroxide (1.20 A/F), suggesting that NiCo hydroxide indeed exhibited a much higher activity than the Ni hydroxide and Co hydroxide for overall electrolysis coupling GOR with HER.*”

REVIEWER COMMENTS

Reviewer #1 (Remarks to the Author):

The paper is publishable in its current form. I only take issue with the statements that the free energy is computed from the entropy etc. Well, then how is the entropy computed? I suspect that authors made the harmonic approximation for the computation of the Gibbs free energies and the authors just have to state this approximation somewhere in the text.

Reviewer #2 (Remarks to the Author):

I really appreciate for the author's efforts to address all my comments. The authors have updated the manuscript properly with my comments. I think this paper should be now published in Nature Communications journal.

Reviewer #3 (Remarks to the Author):

As I stated in the previous revision, this manuscript will be an excellent contribution to the field. Thus, I strongly advise to publish this paper after revising the issues below. Now, I have only one concern about this work.

Main comment

- Figure 3a. I have an important concern about this experiment. At 1.6V the catalyst is clearly forming O₂ in the absence of glycerol. Then, the negative current could be due to the reduction of the previously formed O₂. Thus, the experiment must be repeated using a lower higher potential and a higher lower potential. Is it not necessary to go until 0.2V to reduce the OOH. In fact, Raman results show that for example 1.4V should be enough to for OOH species and 1V enough to reduce them.

Minor comments

- Another detail about the experiment in figure 3a. Have you controlled the time in each step? I am mainly asking by the time at which the systems remain at OCP?

- In several parts of the paper the author wrote "spectroscope techniques". Please, change to Spectroscopy techniques.

- Supplementary figure 38. Please, change to RHE. Do the same please in page 15, line 349. Unfortunately, due to this, I did find the major issue commented above in my first revision.

Point-to-Point Responses to Referees' Comments and Suggestions

First of all, we truly appreciate the referees for their valuable comments and suggestions, which enormously improve the quality and clarity of this manuscript. All of the comments and suggestions have been taken into account in the revised manuscript as follows.

Response to Reviewer #1

COMMENTS TO AUTHOR:

The paper is publishable in its current form. I only take issue with the statements that the free energy is computed from the entropy etc. Well, then how is the entropy computed? I suspect that authors made the harmonic approximation for the computation of the Gibbs free energies and the authors just have to state this approximation somewhere in the text.

Response: We thank the reviewer for acknowledging the acceptance of our work. As suggested by reviewer, we have added the following statement in the manuscript to clarify the method of calculating the entropy in Gibbs free energy calculation.

Page 45, line 461 in Supplementary Information, changed from “*Moreover, calculated Gibbs free energies were referenced to glycerol, glyceric acid, glycolic acid, and formic acid at 1 atm of fugacity.*”

To “*Moreover, calculated Gibbs free energies were referenced to glycerol, glyceric acid, glycolic acid, and formic acid at 1 atm of fugacity, and the entropy and vibrational frequency were calculated using harmonic approximation.*”

Response to Reviewer #2

COMMENTS TO AUTHOR:

I really appreciate for the author's efforts to address all my comments. The authors have updated the manuscript properly with my comments. I think this paper should be now published in Nature Communications journal.

Response: We thank the reviewer for acknowledging the contribution of this work and for the acceptance of our work. We truly appreciate the reviewer for his/her valuable comments and suggestions, which enormously improve the quality and clarity of this manuscript.

Response to Reviewer #3

COMMENTS TO AUTHOR:

As I stated in the previous revision, this manuscript will be an excellent contribution to the field. Thus, I strongly advise to publish this paper after revising the issues below. Now, I have only one concern about this work.

Response: We thank the reviewer for acknowledging the importance of our work. All the concerns raised from the reviewer have been addressed in detail as follows.

Main comment

- Figure 3a. I have an important concern about this experiment. At 1.6V catalyst is clearly forming O₂ in the absence of glycerol. Then, the negative current could be due to the reduction of the previously formed O₂. Thus, the experiment must be repeated using a lower higher potential and a higher lower potential. Is it not necessary to go until 0.2V to reduce the OOH. In fact, Raman results show that for example 1.4V should be enough to for OOH species and 1V enough to reduce them.

Response: We thanks the reviewer for the valuable suggestion. As suggested by the reviewer, we have repeated the intermittent GOR measurement by using a lower oxidation potential and a higher reduction potential. As shown in **Figure A1 and Figure A2**, an anodic potential of 1.4 V (vs. RHE) was first applied on catalysts in KOH electrolyte to form -OOH species. The applied anodic potential was then withdrawn and the glycerol was injected during the open circuit state. Subsequently, a cathodic potential of 1.0 V (vs. RHE) was applied on the catalysts and we observed no reduction current (solid line). By contrast, a large reduction current was observed when no glycerol was added during the open circuit state (dash line). This result is consistent with the previous ones performed at a potential of 1.624 V and 0.224 V. Therefore, we can draw a conclusion that the glycerol dehydrogenation reaction on Ni, NiCo and Co

oxyhydroxide is spontaneous. We have replaced the previous data with the newly added ones which did not involve the potential formation of O₂ and oxygen reduction reaction.

Figure A1. Multi-potential step curves of NiCo hydroxide.

Figure A2. Multi-potential step curves. **a**, Co hydroxide. **b**, Ni hydroxide.

Figure A1 has been substituted for **Fig. 3a** in manuscript.

Figure A2 has been substituted for **Supplementary Fig. 31** in Supplementary Information.

Page 11, line 241 in manuscript, changed “An oxidation current was observed when an anodic potential of 1.624 V (vs. RHE) was first applied on NiCo hydroxide with 1.0 M KOH as the electrolyte.”

To “An oxidation current was observed when an anodic potential of 1.4 V (vs. RHE) was first applied on NiCo hydroxide with 1.0 M KOH as the electrolyte.”

Page 11, line 246 in manuscript, changed “Subsequently, a cathodic potential of 0.224 V (vs. RHE) was applied on the catalyst”

To “Subsequently, a cathodic potential of 1.0 V (vs. RHE) was applied on the catalyst”

Minor comments

- Another detail about the experiment in figure 3a. Have you controlled the time in each step? I am mainly asking by the time at which the systems remain at OCP?

Response: We thank the reviewer for the valuable question. The time in each step (including the oxidation step, OCP step, and reduction step) have been controlled to be 2 min. To show the experimental procedure more clearly, we have marked the exact time in **Fig. 3a** in manuscript and **Supplementary Fig. 31** in Supplementary Information.

- In several parts of the paper the author wrote “spectroscope techniques”. Please, change to Spectroscopy techniques.

Response: We thank the reviewer for the valuable comments. We have corrected all the “spectroscope techniques” in manuscript to “spectroscopy techniques”.

Page 1, line 25 in manuscript, changed “*spectroscope techniques*” to “*spectroscopy techniques*”.

Page 3, line 75 in manuscript, changed “spectroscope techniques” to “*spectroscopy techniques*”.

- Supplementary figure 31. Please, change to RHE. Do the same please in page 15, line 349. Unfortunately, due to this, I did find the major issue commented above in my first revision.

Response: We thank the reviewer for the valuable comment. As suggested by reviewer, the potentials in Supplementary Fig. 31 have been converted to be the ones versus reversible hydrogen electrode (RHE) as shown in **Figure A2**.

Figure A2 has been substituted for **Supplementary Fig. 31** in Supplementary Information.

Page 15, line 349 in manuscript, changed “*To confirm these changes in electron filling predicted by DFT, we carried out soft X-ray absorption spectroscopy (sXAS) measurements on the sample after operating at 0.6 V (vs. Ag/AgCl) in KOH with/without glycerol, which represent the sample states with/without oxygen vacancy.*” To “*To confirm these changes in electron filling predicted by DFT, we carried out soft X-ray absorption spectroscopy (sXAS) measurements on the sample after operating at 1.624 V (vs. RHE) in KOH with/without glycerol, which represent the sample states with/without oxygen vacancy.*”